# Deep Counterfactual Estimation with Categorical Background Variables

**Edward De Brouwer**
ESAT-STADIUS
KU Leuven
edward.debrouwer@esat.kuleuven.be

## Abstract

Referred to as the third rung of the causal inference ladder, counterfactual queries typically ask the "*What if ?*" question retrospectively. The standard approach to estimate counterfactuals resides in using a structural equation model that accurately reflects the underlying data generating process. However, such models are seldom available in practice and one usually wishes to infer them from observational data alone. Unfortunately, the correct structural equation model is in general not identifiable from the observed factual distribution. Nevertheless, in this work, we show that under the assumption that the main latent contributors to the treatment responses are categorical, the counterfactuals can be still reliably predicted. Building upon this assumption, we introduce CounterFactual Query Prediction (CFQP), a novel method to infer counterfactuals from continuous observations when the background variables are categorical. We show that our method significantly outperforms previously available deep-learning-based counterfactual methods, both theoretically and empirically on time series and image data. Our code is available at https://github.com/edebrouwer/cfqp.

## 1 Introduction

Counterfactual queries aim at inferring the impact of a treatment conditioned on another observed treatment outcome. Typically, given an individual, a treatment assignment, and a treatment outcome, the counterfactual question asks what would have happened to that individual, had it been given another treatment, everything else being equal. An illustrative and motivating example is the case of clinical time series. Based on the observation of the outcome of treatment $A$ on a particular patient, counterfactual queries ask what would have been the outcome for this patient, had it been given treatment $B$ instead. Notably, counterfactual prediction differs from interventional prediction, which is also referred to as counterfactual potential outcomes [23] and constitutes the second rung of the causation ladder [19]. Counterfactual predictions are retrospective, as they condition on an observed treatment outcome. In contrast, interventional predictions are prospective as they only condition on observations obtained before treatment assignment.

Much more than a statistical curiosity, counterfactual reasoning reflects complex cognitive abilities that are deeply ingrained in the human brain [22] and emerges in the early stages of cognitive development [7]. The ability to reason counterfactually can indeed help to identify causes of outcomes retrospectively, has been suggested to be central in the formation of rational intention, and supports key theories of human cognition [28]. The importance of counterfactual reasoning in the human cognitive process has thus motivated researchers to endow artificially intelligent systems with the same ability [21]. However, counterfactual inference is not possible from observational and interventional data alone [20].

36th Conference on Neural Information Processing Systems (NeurIPS 2022).

Counterfactual reasoning, therefore, requires making several assumptions to overcome this limitation. The most popular one assumes knowledge of the underlying structural equation model that describes the data generating process [21] or a specific functional form thereof [1, 5, 21]. Unfortunately, this assumption is rarely met in practice, especially in high-dimensional data such as time series or images. This led to the development of *deep* structural equation models that attempt to model the structural equations with neural networks [17, 24]. However, despite their ability to model high-dimensional data, these approaches fail to provide theoretical guarantees for the reconstruction of counterfactuals. Indeed, they focus on modeling the factual distribution which, without further assumption, can, unfortunately, lead to erroneous counterfactual distributions.

In this work, we bridge the gap between the classical structural equation model assumptions and deep-learning-based architectures. By assuming that the treatment and observables are continuous and that the hidden variables that contribute most to the treatment response are categorical, we can rely on recent results in identifiability of mixture distributions [3] to show that we can approximately recover the counterfactuals using arbitrary parametric functions (*i.e.* deep neural networks) to model the causal dependence between variables. This allows us to infer counterfactuals on high-dimensional data such as time series and images. Generally, this work explores the assumptions that can lead to approximate counterfactual reconstructions while controlling the discrepancy between the recovered and the true counterfactual distributions.

Besides the general appeal of endowing machine learning architectures with counterfactual reasoning abilities, an important motivation for our work is the counterfactual estimation of treatment effects in clinical patient trajectories. In this motivational example, one wishes to predict the individual treatment effect retrospectively. Based on the observed treatment outcome of a particular patient, we want to predict what would have been the outcome under a different treatment assignment. The ability to perform counterfactual inference on patient trajectories has indeed been identified as a potential tool for improving long and costly randomized clinical trials [15].

**Contributions**

- We provide a new set of assumptions under which the counterfactuals are identifiable using arbitrary neural networks architecture, bridging the gap between structural equation models and deep learning architectures.

- We derive a new counterfactual identifiability result that motivates a novel counterfactual reconstruction architecture.

- We evaluate our construction on three different datasets with different high-dimensional modalities (images and time series) and demonstrate accurate counterfactual estimation.

## 2 Background

### 2.1 Problem Setup : Counterfactual Estimation

We consider the general causal model $M = \langle U, V, F \rangle$ depicted in Figure 1a consisting of background variables $U$, endogenous variables $X, T$ and $Y$ and the set of structural functions $F$. Background variables $U = \{U_X, U_T, U_\epsilon, W\}$ are hidden exogenous random variables that determine the values of the observed variables $V = \{X, T, Y\}$. Covariates $X \in \mathcal{X}$ represent the information available before treatment assignment, $T \in \mathcal{T}$ is the treatment assignment and $Y \in \mathcal{Y}$ is the observed response to the treatment. We refer to the space of probability measures on $\mathcal{Y}$ as $\mathcal{P}(\mathcal{Y})$. Observed variables $V$ are generated following the structural equations $F = \{f_X, f_T, f_Y, f_\epsilon\}$, such that $X = f_X(U_x, W)$, $T = f_T(U_T, X)$, $U_\epsilon = f_\epsilon(W)$, $Y = f_Y(X, T, U_\epsilon)$. We further assume *strong ignorability* (*i.e.* no hidden confounders between $T$ and $Y$).

Using notations introduced in Pearl et al. [21], we define the potential response of a variable $Y$ to an action $do(T = t)$ for a particular realization of $U = u$ as $Y_t(u)$. Our goal is to predict the counterfactual response, for a new treatment assignment ($T = t'$), conditioned on an observed initial treatment response. That is, the probability of observing a different treatment response under treatment $t'$, after observing treatment response $y$ for covariate $x$ and treatment $t$. The probability density function of counterfactual $y'$ then writes:

$$p(Y_{t'} = y' \mid X = x, Y = y, T = t) = \frac{p(Y_{t'} = y', X = x, T = t, Y = y)}{p(X = x, T = t, Y = y)}$$

$$= \int_u p(Y_{t'}(u) = y')p(U = u \mid X = x, T = t, Y = y), \quad (1)$$

and we refer to the counterfactual probability measure as $\nu_{t'}(x, y, t)$. Equation 1 suggests a natural three step procedure for computing the probability of counterfactual. First, the *abduction* step infers the density of $U$ conditioned on the observed treatment outcomes, covariates and treatments: $p(U = u \mid X = x, T = t, Y = y)$. Second, in the *action* step, one sets the new treatment in the causal model ($do(T = t')$). Lastly, in the *prediction* step, one can propagate the values of $U = u$ and $T = t'$ in the causal graph, using $F$, to compute $p(Y_{t'}(u) = y')$.

In practice, we only have access to a set of $N$ observations of variables $X$, $Y$ and $T$. We refer to this dataset as $\mathcal{D} = (\mathbf{X}, \mathbf{T}, \mathbf{Y})$ where $\mathbf{X} = \{x_i : i = 1, ..., N\}$, $\mathbf{Y} = \{y_i : i = 1, ..., n\}$ and $\mathbf{T} = \{t_i : i = 1, ..., N\}$. Importantly, we don't have access to counterfactual examples (*i.e.* a tuple $(x, y, t, t', y')$), such that direclty learning a map $(x, y, t, t') \to y_{t'}$ is excluded.

## 2.2 General Non-identifiability of Counterfactuals

Because the background variables $U$ are hidden, the above three-steps procedure requires knowledge of the structural functions $F = \{f_X, f_T, f_Y\}$. Indeed, one can show that there exist multiple structural functions $F$ that would lead to the same observed joint density $p(X, Y, T)$ but would lead to incorrect counterfactual probabilities [19, 20]. The correct causal model is thus in general non-identifiable, leading to non-identifiability of the counterfactual probability. We specify what is meant by the identifiability of counterfactuals in the following definition.

**Definition 1** (Identifiability of Counterfactuals). Let $\rho$ be a metric on $\mathcal{P}(\mathcal{Y})$, $\nu_{t'}(X, Y, T)$ the true counterfactual probability measure and $\hat{\nu}_{t'}(X, Y, T)$ the estimator of the counterfactual probability measure with $N$ data points. Counterfactuals are $\rho$-identifiable at threshold $\delta$ if, for all $t, t' \in \mathcal{T}, x \in \mathcal{X}, y \in \mathcal{Y}$,

$$\lim_{N \to \infty} \rho(\nu_{t'}(x, y, y), \hat{\nu}_{t'}(x, y, t)) \leq \delta$$

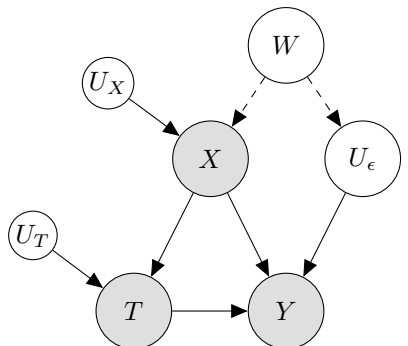

(a) General Bayesian network for the treatment counterfactual problem. $X$, $Y$ and $T$ are observed while $U_X, U_T, W$ and $U_\epsilon$ are hidden background variables.

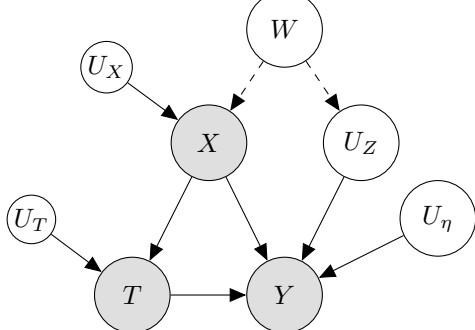

(b) Bayesian Network embodying the hidden categorical background variable assumption. $U_\epsilon$ is split in a background categorical variable $U_Z$ and a continuous background variable $U_\eta$.

Figure 1: Graphical model representations of the causal model $M$. We assume strong ignorability and continuous treatments $T$, observables $X$ and responses $Y$.

## 2.3 Causal Model Assumptions for Counterfactual Idenfiability

Despite the general non-identifiability of structural equation models laid out above, we propose plausible assumptions that one can build upon to identify counterfactuals reliably. We first assume $X$ and $Y$ are continuous (potentially high dimensional) variables (such as images or time series). The treatment assignment $T$ is also assumed continuous, and $\mathcal{X} \times \mathcal{T}$ is a connected space. Our first central assumption posits that the hidden variable $U_\epsilon$ factorizes into a categorical and a continuous variable.

**Assumption 1** (Categorical Background Variable). The background variable $U_\epsilon$ decomposes into a categorical latent variable $U_Z \in [K] = \{1, .., K\}$ and an independent exogenous continuous variable $U_\eta$.

This assumption is depicted in the graphical model of Figure 1b and embodies the intuition of different hidden groups that drive the treatment response. For instance, a treatment could have different responses depending on the stage of the disease a patient finds themself in. The disease stage is unobserved yet correlated with the observed covariates $X$ (through $W$).

Due to the categorical nature of variable $U_Z$, one can write the conditional density of $Y$ as a mixture model:

$$p(Y = y \mid X = x, T = t) = \sum_{u_Z \in \{1,..,K\}} P(U_Z = u_Z) \cdot \int p(U_\eta = u_\eta) \mathbb{I}[f_Y(x, t, u_Z, u_\eta) = y] du_\eta$$

We define $\gamma$ as the probability density function of the conditional treatment response generated by $f_Y$, $U_Z$ and $U_\eta$. $\gamma$ is thus a mixture probability density function with mixture components[1] $\gamma_k \in \mathcal{P}(\mathcal{Y})$ and weights $\omega_k$.

$$Y \mid X, T \sim \gamma(X, T) = \sum_{k=1}^{K} \omega_k \gamma_k(X, T) \tag{2}$$

Without loss of generality, we assume that $U_\eta \sim \mathcal{N}(\mathbf{0}, \Sigma^2)$. In the case of additive noise, the conditional distribution of Y becomes a mixture of Gaussians: $\gamma(X, T) = \sum_{k=1}^{K} \omega_k \mathcal{N}(\mu_k(X, T), \Sigma_k^2)$, where $\mu_k$ are functions mapping $X$ and $T$ to the mean of the mixture components and we consider different variances for each $k$. We now proceed with the next assumptions.

**Assumption 2** (Continuity). The moments of the probability density functions $\gamma_k(x, t)$ exist and are continuous functions of $X$ and $T$:

$$\mu_k^r(x, t) = \mathbb{E}_{Y \sim \gamma_k(x,t)}[Y^r] \in C(x, t) \quad \forall r \in \mathbb{N}, k \in [K] \tag{3}$$

**Assumption 3** (Clusterability). For each $(x, t) \in (\mathcal{X}, \mathcal{T})$, the density $\gamma(x, t)$ is clusterable and the expected deviation of each $\gamma_k(x, t)$ is bounded by a constant $\delta \in \mathbb{R}$. That is, $\forall k \in [K], \forall x, t \in (\mathcal{X} \times \mathcal{T})$, with $\mu_k(x, t) = \mathbb{E}_{Y \sim \gamma_k(x,t)}[Y]$:

$$\mathbb{E}_{Y \sim \gamma_k(x,t)}\big[\|Y - \mu_k(x, t)\|_2\big] \leq \delta \tag{4}$$

In our motivating clinical example, Assumption 2 reflects that the probability of specific treatment response changes continuously over the set of observed covariates and treatments. In particular, the expected treatment outcome for a particular patient varies continuously with the treatment assignment, which is a common assumption, *e.g.* in clinical practice [14].

---

[1] The mixture components are defined such that for any subset $\mathcal{A} \subset \mathcal{Y}$, we have $\int_{\mathcal{A}} \gamma_k(X, T)(y) dy = \int_{\infty}^{\infty} \mathbb{I}[f_Y(X, T, U_Z = k, U_\eta) \in \mathcal{A}] dP(U_\eta)$. The mixutre weights are defined as $\alpha_k = P(U_Z = k)$.

Assumption 3 posits *clusterability* of the mixture components $\gamma_k$ for which a rigorous mathematical definition is given in Appendix C. It is motivated by recent results on the identifiability of mixtures models[3]. Intuitively, it supposes that patients with the same observed covariates and treatment assignment but different hidden group will show different treatment outcomes. We also bound the expected deviation of the mixture components $\gamma_k$ that characterize the inter-group variability in the treatment response for a particular patient and treatment outcome.

## 3  Methods

### 3.1  Identifiability and Counterfactuals Reconstruction

For a fixed point $(X = x, T = t)$, Equation 2 is a finite mixture model, for which identifiability results are available [3]. Notably, these results guarantee identifiability up to a permutation of the latent class assignment $\sigma(\cdot) : [K] \to [K]$. That is, there exists some permutation $\sigma(\cdot) : [K] \to [K]$ such that $\hat{\gamma}_{\sigma_k(k)(x,t)} \approx \gamma_k(x,t)$, $\hat{\omega}_{\sigma_k(k)} \approx \omega_k$, where $\hat{\gamma}$ and $\hat{\omega}$ are the estimated density functions and weights. However, it does not entail identifiability of the counterfactuals in the sense of definition 1. Indeed, the action step of the counterfactual strategy from Section 2.1 requires a consistent permutation $\sigma$ across the whole domain $(\mathcal{X} \times \mathcal{T})$ in order to reuse the inferred class assignments $\hat{U}_Z$ at a specific point $(X = x, T = t)$ to predict the counterfactual at another point $(X = x, T = t')$ — with a different treatment assignment. Nevertheless, using the assumptions from the previous section, we can still ensure the identifiability of the counterfactuals as the following result confirms:

**Result 3.1** (Identifiabilty of Counterfactuals with Categorical Background Variables). *Let X, T and Y be continuous random variables generated according to the graphical model of Figure 1b with the domain of X and T being connected. Let $W_1(\cdot, \cdot)$ be the first Wasserstein distance on $\mathcal{P}(\mathcal{Y})$, $\nu_{t'}(X, Y, T)$ the probability distribution of $Y_{t'} \mid X, Y, T$ and $\hat{\nu}_t^N(X, Y, T)$ its estimator from N observed data points. If Assumptions 1, 2 and 3 hold, for each $(x,t)$, the counterfactual distribution is $W_1$-identifiable in expectation at threshold $\delta$:*

$$\lim_{N \to \infty} \mathbb{E}_{Y \sim \gamma(x,t)} \left[ W_1(\nu_{t'}(x, Y, t), \hat{\nu}_{t'}^N(x, Y, t)) \right] \leq \delta$$

*In the special case when the noise response is additive, we have*

$$\lim_{N \to \infty} W_1(\nu_{t'}(X, Y, T), \hat{\nu}_{t'}^N(X, Y, T)) = 0$$

The proof is given in Appendix C. This result gives us a bound on the distance between the inferred and true counterfactual distributions in the asymptotic regime. Importantly, it does not restrict the dimension of $\mathcal{X}$ and $\mathcal{Y}$, and is thus valid on challenging data modalities such as time series or images.

**Continuity of distribution and complexity**   The result above holds asymptotically in the number of available samples. In the additive Gaussian case, the sample complexity for learning a $K$-mixture model with $Y \in \mathbb{R}^d$ within $\epsilon$ total variation distance is $\tilde{O}(Kd^2/\epsilon)$ [4]. Fortunately, the continuity assumption (Assumption 2) saves us from having to learn an individual mixture at each point $(X = x, T = t)$, by jointly learning the continuous moments functions $\mu_r'(X, T)$. A better sample complexity bound can then be derived with further assumptions on $\mu_r'(X, T)$.

### 3.2  CFQP : CounterFactual Query Prediction

Equipped with those theoretical results, we introduce CFQP, a counterfactual prediction model based on a neural Expectation-Maximization mechanism. The basic building block of CFQP is a base-model $m(x, t)$, that predicts the treatment response $y$ based on covariates and treatment assignment. For each latent category $k$, we learn a base-model that approximates the individual treatment response in that category : $m_k(x, t) \approx \mu_k(x, t)$. Our theoretical results require the true number of classes of $K_0$ to be known in advance. Yet, this is rarely the case in practice, and we describe our architecture for an arbitrary number of classes $K$. The learning of the base-models follows three steps: a joint initialization, an expectation phase, and a maximization phase. The overall process is depicted in Figure 2. We also present a pseudo-code description of the procedure in Algorithm 1.

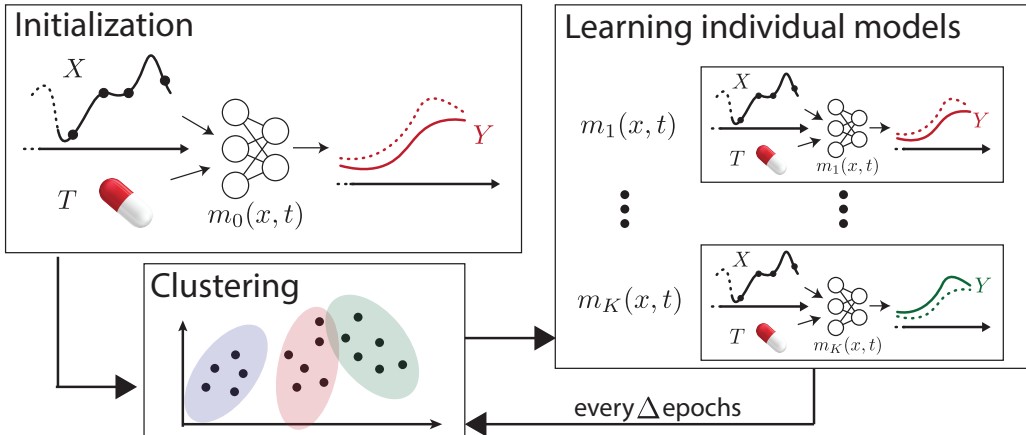

Figure 2: Training procedure of CFQP. We first initialize a common model $m_0$. Second, we cluster the different available data points into $K$ clusters. We then use this clustering to train $K$ individual models (Maximization step). The clustering is updated every $\Delta$ epochs (Expectation step).

**Initialization** We first train a single common base-model $m_0$ on all data points $\mathbf{X}, \mathbf{T}$ and $\mathbf{Y}$. Because it discards the variability of the hidden variable $U_Z$, this model will approximate the conditional average treatment response : $m_0(x, t) \approx \mathbb{E}[Y \mid X = x, T = t]$.

**Expectation - Clustering** The clustering step ensures that each base-model $m_k$ is trained with the data from the corresponding latent category. We distinguish between an *initial* clustering and an *update* clustering stage.

The *initial* clustering assignment happens after the initialization step has converged. We then use the clusterability assumption (Assumption 3) to drive the assignment and use $K$-means clustering based on the residuals $\|y - m_0(x, t)\|_2^2$.

The *update* clustering step happens at regular intervals to improve the quality of the cluster assignment as the performance of each individual base-model increases. A point $(x, y, t)$ is assigned to cluster $k = \underset{j}{\arg\min} \|y - m_j(x, t)\|_2^2$.

**Maximization - Learning individual base-models** Once the dataset $\mathcal{D} = (\mathbf{X}, \mathbf{T}, \mathbf{Y})$ is clustered in $K$ groups, we train the individual models $m_k$ on the corresponding cluster. This corresponds to the maximization step of the expectation-maximization scheme. Every $\Delta$ epochs, we update the cluster assignment until convergence.

**Counterfactual prediction** For a data point $(x, t, y)$, we first infer the cluster assignment $k = \underset{j}{\arg\min} \|y - m_j(x, t)\|_2^2$. We predict the counterfactual for treatment $t'$ as $\hat{y}' = m_k(x, t')$.

## 4 Related Work

Causal perspectives in machine learning have gained significant traction in the past years [26, 25]. Among them, one distinguishes between causal *discovery* approaches, aiming at discovering the causal relations between variables [10, 9], and causal *inference*, aiming at building treatment effects estimators from data [13]. Our work belongs to the latter. A common way to estimate counterfactuals is to posit a specific structural causal model [20] or its specific functional form [5], therefore ensuring identifiability. Synthetic controls are an example of this strategy, assuming an underlying linear structure [1]. However, these approaches are by definition restrictive in terms of the expressivity of the structural equations. Deep learning approaches for counterfactual estimation have been recently proposed to address this issue [17, 24] but without identifiability guarantee. For the discrete case, assumptions have been proposed to bridge this gap, such as monotonicity or generalization

---

**Algorithm 1:** CFQP Training

---

**Data:** $\mathbf{X}, \mathbf{Y}, \mathbf{T}$, the number of latent clusters $K$, a number of epochs $e_{max}$, the update-period $\Delta$.
**Result:** A list of $K$ hidden models $m_k$.
Initialize a single base model $m_0$ at random.
**for** *epoch* $\leftarrow 0$ **to** $e_{max}$ **do**
   Train $m_0$ minimizing $\mathcal{L}_0 = \mathbb{E}\left[(m_0(X, T) - Y)^2\right]$
**end**
Compute residuals $\mathbf{r} = (m_0(\mathbf{X}, \mathbf{T}) - \mathbf{Y})$.
Assign the residuals of each training sample into $K'$ clusters (initial).
Initialize each $m_k$ with $m_0$.
**for** *epoch* $\leftarrow 0$ **to** $e_{max}$ **do**
   **for** $i \leftarrow 0$ **to** $\Delta$ **do**
      Train $m_k$ minimizing $\mathcal{L}_k = \mathbb{E}\left[(m_k(X_k, T_k) - Y_k)^2\right]$ for each cluster.
   **end**
   Compute residuals $\mathcal{R} = \{(m_k(\mathbf{X}, \mathbf{T}) - \mathbf{Y}) : k = 1, ..., K\}$.
   Assign the residuals of each training sample into $K$ clusters (update).
**end**
**Return** trained models $m_k$.

---

thereof [16, 13]. Nevertheless, because they focus on addressing the discrete case, they are not directly applicable to high-dimensional data such as time series, which is a motivation for our work.

Our work builds upon the literature on the identifiability of finite mixture distributions where guarantees have been achieved in the Gaussian [8] and non-parametric cases [3], among others.

## 5 Experiments

### 5.1 Datasets

Treatment outcomes are often complex and high-dimensional. We thus evaluate our proposed model architecture on high-dimensional data modalities: time series and images. As counterfactual inference evaluation on real-world data is a complex and ongoing research area, we use synthetic datasets inspired by case studies from the literature. Details about the data generation are given in Appendix A.

#### 5.1.1 Counterfactual Image Transformation

To explore the capacity of our model to operate on images, we generate a dataset inspired by [17], building upon modification of MNIST images. Covariates $X$ are original MNIST images and the treatment $T$ is a spatial rotation applied to the image. $U_Z$ is a coloring of the image and the treatment outcome $Y$ is the resulting colored and rotated MNIST image. $U_\eta$ is either an additive Gaussian noise at each pixel or a Gaussian blur with a random kernel.

#### 5.1.2 Counterfactual Longitudinal Treatment Effect Prediction

Next, we explore the performance of our model on time series data. As per our motivational example, our goal is here to infer counterfactuals from clinical trajectories with a hidden patient class. $X$ and $Y$ are multi-dimensional time series, $U_Z$ is a latent patient group and $T$ is a continuous treatment assignment. We used two different datasets: an harmonic oscillator and a cardiovascular model.

**Harmonic oscillator.** We simulate the angular positions of two coupled harmonic oscillators. The treatment consists of applying different gradual offsets to the time series. We consider three hidden groups ($K_0 = 3$) that modulate the treatment response differently.

**Cardiovascular simulator.** We use a simulator of the cardiovascular system proposed in [29] to predict the impact of fluid intake on blood pressure. Fluids are commonly administered in intensive care for treating severe hypotension. However, the individual patient response is difficult to assess a priori, as it depends on clinically hidden variables. We consider here two hidden groups of patients

with distinct responses to fluid intake. $X$ and $Y$ contain the arterial and venous blood pressure time series before and after fluid intake. The treatment is the amount of fluid injected. We add process noise by either considering an additive Gaussian noise on the observed treatment response or by introducing noise in the fluid injection process, leading to a non-linear and complex perturbation.

## 5.2 Baselines

We compare our approach against multiple counterfactual inference models from the literature: *Diff-SCM* [24], a recently introduced diffusion based counterfactual model; *Deep-SCM* [17], a deep counterfactual inference architecture building on normalizing flows ; Synthetic Controls (*SC*) [1], a well known linear method for counterfactuals estimation and *Deep-ITE*, a deep individual treatment effect estimator such as proposed in [27, 12], using direct predictions from $X$ and $T$.

Table 1: Test MSE of the counterfactual reconstructions for the different datasets.

| | Additive Noise | | | Non-Additive Noise | | |
|---|---|---|---|---|---|---|
| Model | Harmonic Oscillator | colored-MNIST | Cardiovascular | Harmonic Oscillator | colored-MNIST | Cardiovascular |
| Deep-ITE [12] | $0.187 \pm 0.006$ | $0.017 \pm 0.001$ | $1.084 \pm 0.087$ | $0.174 \pm 0.004$ | $0.017 \pm 0.001$ | $1.14 \pm 0.121$ |
| SC [1] | $0.177 \pm 0.131$ | $0.020 \pm 0.001$ | $1.610 \pm 0.141$ | $0.167 \pm 0.004$ | $0.020 \pm 0.001$ | $1.628 \pm 0.144$ |
| Deep-SCM [17] | $0.124 \pm 0.005$ | $0.011 \pm 0.001$ | $0.405 \pm 0.042$ | $0.123 \pm 0.005$ | $0.011 \pm 0.001$ | $0.424 \pm 0.042$ |
| Diff-SCM [24] | $0.082 \pm 0.023$ | $0.008 \pm 0.004$ | $0.206 \pm 0.036$ | $0.106 \pm 0.038$ | $0.009 \pm 0.002$ | $0.311 \pm 0.073$ |
| CFQP (ours) | $\mathbf{0.013 \pm 0.001}$ | $\mathbf{0.001 \pm 0.001}$ | $\mathbf{0.077 \pm 0.050}$ | $\mathbf{0.009 \pm 0.001}$ | $\mathbf{0.002 \pm 0.001}$ | $\mathbf{0.188 \pm 0.114}$ |

## 5.3 Counterfactuals Prediction

We evaluate our approach in terms of the quality of the counterfactual reconstructions. We generate counterfactuals $(t',y')$ for each observation $(x,y,t)$ in the test set, resulting in a tuple $(x,y,t,t',y')$. We evaluate the MSE between the true counterfactual $y'$ and the estimated one $\hat{y}'$ as $\text{MSE}_{cf} = \|y' - \hat{y}'\|_2^2$. We set the number of clusters $K$ as an hyper-parameter and select the value that result in lowest validation error. The results are presented in Table 1. We use additive and non-additive noise variants for each dataset. For each dataset and each variant, we see that our approach outperforms the other baselines, demonstrating its experimental effectiveness.

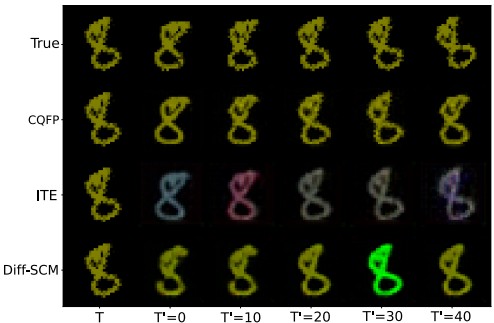

Figure 3: Example of reconstructed counterfactuals on the image dataset for different methods (best seen in color). Here, $U_Z$ corresponds to the color of the digit and $T$ to the rotation angle. $X$ is the non-rotated and non-colored image while $Y_T$ is the colored and rotated image. First column is the observed factual treatment outcome. Subsequent columns show the counterfactual reconstructions for different values of $T'$. First row is the ground truth.

In Figure 3, we also report examples of counterfactual reconstructions on the image dataset. Each row corresponds to counterfactual estimations for a particular method (top is the truth), and each column represents a different treatment assignment $T'$ (the left column is the factual $T$). Because it cannot model the categorical latent variable, the Deep-ITE model gets the expected individual treatment effect right but fails to accurately estimate the counterfactuals. For Diff-SCM, we observe that, despite providing an accurate reconstruction for most of the treatment assignments, the lack of guarantees on the deep structural equation model leads to potential incorrect reconstructions (*e.g.* at $T' = 30$). To assess the quality of the counterfactual images reconstructions, we also evaluate the structural similarity index and report the results in Appendix A.1.1. We observe that CFQP outperforms all baselines on that metric.

## 5.4 Robustness Analysis

We further complement our experimental section with a comprehensive robustness analysis. We first investigate the impact of misspecifying the number of latent classes (*i.e.* when $K_0 \neq K$). We then investigate the impact of a correlation between $X$ and $Z$ with increasing strength. Finally, we investigate the robustness of our approach to different noise responses.

### 5.4.1 Number of latent classes

We study the evolution of performance when the number of classes $K$ is misspecified ($K_0 \neq K$). In Figure 4a, we show the reconstruction error on the validation set (*i.e.*, on the factual data) and the counterfactual reconstruction error of CFQP for different values of $K$ on the image dataset with true number of latent classes $K_0 = 6$. The validation MSE is lowest for $K = 6$ and $K = 7$, which corresponds to optimal test reconstruction error. This hyper-parameter can thus easily be tuned by monitoring the treatment prediction performance on the factual data. In Appendix B.2, we further report the quantitative results for the other datasets in function of the number of the clusters $K$.

### 5.4.2 Correlation between $X$ and $U_Z$

Our approach holds even when the latent background variable $U_Z$ and the covariates $X$ are correlated. When this is the case, information from $X$ can be used to infer the value of $U_Z$. In Figure 4b, we study the impact of the correlation strength $\rho$ between $X$ and $U_Z$. More details about the definition of $\rho$ are to be found in Appendix B. We compare the performance of CFQP with a Deep-ITE model. We observe that the counterfactual performance of our model remains constant, regardless of the correlation strength. However, as the correlation becomes more important, $U_Z$ becomes a deterministic outcome of $X$ and Deep-ITE eventually converges to the performance of CFQP.

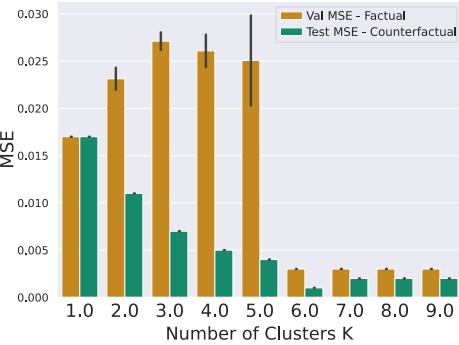
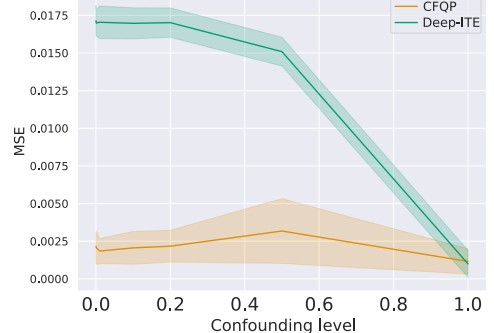

(a) Reconstruction MSE in function of the number of groups. The true number of groups is 6.

(b) Evolution of the reconstruction MSE in function of the correlation strength between $X$ and $U_Z$.

Figure 4: Robustness analysis for the number of latent classes (left) and correlation strength (right).

### 5.4.3 Non-additive Noise Responses

Our model architecture is motivated by the additive noise case. Yet, our theoretical results hold for arbitrary distributions. In Table 1, we reports the performance of the different models under non-additive noise responses. We observe an overall small degradation of performance of all methods but CFQP still outperforms baselines, as predicted by our identifiability results.

Table 2: Fluids PEHE

| Model | Cardiovascular PEHE ↓ |
|---|---|
| Deep-ITE [12] | $0.594 \pm 0.160$ |
| SC [1] | $0.258 \pm 0.016$ |
| Diff-SCM [24] | $0.377 \pm 0.049$ |
| CFQP (ours) | $\mathbf{0.143 \pm 0.108}$ |

## 5.5 Retrospective Individualized Treatment Effect Estimation

We explore the ability of our model to infer individualized treatment effects retrospectively. Based on observed treatment outcomes, we predict the difference in treatment response between two treatment

regimes for a single individual. The metric of interest is the Precision in Estimation of Heterogeneous Effects PEHE [11]. For two treatment regimes $t'$ and $t''$, we write

$$\text{PEHE}(t', t'') = \sqrt{\mathbb{E}_{X,T,Y}\left[\left((Y_{t''} - Y_{t'}) - (\hat{Y}_{t''} - \hat{Y}_{t'})\right)^2 \mid X, Y, T\right]}. \tag{5}$$

In Table 2, we report $\text{PEHE}(t' = 0.5, t'' = 0.8)$ for test patients in the Cardiovascular dataset with different fluid intake. We observe that the methods leveraging information about treatment outcomes, CFQP, SC and Diff-SCM outperform the classical individual treatment effect estimators (Deep-ITE).

### 5.5.1 Impact of the clustering algorithm

Our approach relies on an *initial* clustering step as described in Section 3.2. Different clustering strategies can be considered for this step. In Appendix B.5, we investigate the difference between K-means and Gaussian mixture models. Only minor deviations are observed in terms of performance between both approaches.

## 6 Conclusion

Estimating counterfactuals from observational studies is one of the most challenging tasks in causal inference. In this work, we proposed a set of reasonable assumptions that allow computing counterfactuals on high-dimensional data while harnessing the power of modern machine learning architectures. Based on these assumptions, we derived a new counterfactual model and demonstrated favorable experimental performance. In particular, we showed that this approach could be used to infer individual treatment effects a posteriori in clinical patient trajectories. Nevertheless, the set of assumptions proposed here is not unique, and depending on the specific applications, others might be deemed more relevant. Indeed, the space of assumptions that allow to approximately recover counterfactuals with a controlled level of error is a potentially very rich research direction. Other model architectures compatible with our set of assumptions are also possible.

**Potential negative societal impacts** Counterfactual inference has been used in clinical settings and for assessing the impact of public policies [2], among others. However, as shown in this paper, its correctness hinges on assumptions whose validity is not always met, potentially leading to misleading conclusions.

**Acknowledgements** Edward is funded by a FWO-SB grant. Edward would also like to thank Zeshan Hussain, David Sontag and Adam Arany for their comments on preliminary versions of this manuscript.

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
