## A Datasets

### A.1 Image Dataset Generation

The image dataset uses the MNIST handwritten digits images dataset. We use the following data generating process:

- $X \in \{0, ..., 255\}^{28 \times 28 \times 1}$ is an MNIST image sampled at random from the dataset. We write $y_{label}$ the label of the digit present in the image.

- $T \in \mathcal{R} = \mathcal{U}(0, 0.3) + 5 \cdot \sigma \left( \frac{\frac{1}{784} \sum_{i=1}^{28} \sum_{j=1}^{28} X_{i,j} - 33}{11} \right)$ with $\sigma(\cdot)$ the sigmoid function. This encodes confounding between the observables $X$ and the treatment assignment $T$.

- $p_0 = \frac{1}{K}$

- $p \in [0, 1]^K$

- $\rho \in [0, 1]$ is the strength of confounding between $X$ and $U_Z$

- $p[i] = \frac{1 - ((1 - p_0)\rho + p_0)}{K - 1} \quad \forall i \in [K], i \neq y_{label} \mathrm{mod} K$

- $p[i] = ((1 - p_0)\rho + p_0) \quad \forall i \in [K], i = y_{label} \mathrm{mod}(K)$

- $U_Z \sim \mathcal{M}\mathrm{ultinomial}(p)$

- $Y \in \{0, ..., 255\}^{28 \times 28 \times 1}$ is the rotated image $X$ with angle $T$ and Gaussian blur with kernel size $5 \times$ and standard deviation $\sigma$.

Examples of generated images $Y$ are shown in Figure 5. We use the original MNIST training set that we randomly divide as $70\%$ training set, $15\%$ validation set and $15\%$ test set.

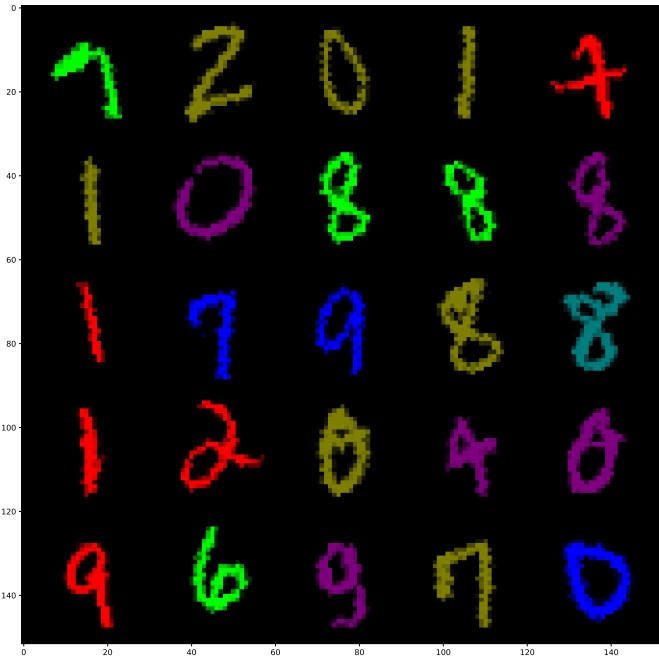

Figure 5: Examples of colored and rotated MNIST images $Y$

### A.1.1 Evaluating the structural similarity index measure of the counterfactual images predictions

Mean square error between the ground truth and predictions might not always be the most meaningful way to assess the quality of the reconstruction. A widely used metric in computer vision is the structural similarity index measure (SSIM). In Table 3, we report the SSIM between the predicted and ground truth images counterfactuals. We observe that CFQPoutperforms all baselines in that metric, producing more accurate counterfactual predictions.

Table 3: Structural similarity results for the MNIST reconstructions

|  | Additive Noise | Non-Additive Noise |
| --- | --- | --- |
| Model | colored-MNIST | colored-MNIST |
| Deep-ITE [12] | $0.490 \pm 0.006$ | $0.484 \pm 0.004$ |
| SC [1] | $0.020 \pm 0.001$ | $0.020 \pm 0.001$ |
| Deep-SCM [17] | $0.857 \pm 0.003$ | $0.856 \pm 0.004$ |
| Diff-SCM [24] | $0.780 \pm 0.063$ | $0.750 \pm 0.052$ |
| CFQP (ours) | $0.964 \pm 0.003$ | $0.959 \pm 0.005$ |

## A.2 Harmonic Oscillator

We consider input times series $X \in \mathbb{R}^{d_x \times t_x}$, with $d_x = 2$ the number of temporal dimensions and $t_x$ the length of the input time series. The temporal responses are given by $Y \in \mathbb{R}^{d_y \times t_y}$. We use the following data generating process :

$$
\begin{aligned}
U_Z &\sim \{0, 1, 2\} \\
T &\sim \mathcal{U}(0.2, 1) \\
t_x &= (0, ..., 19) \\
t_y &= (20, ..., 40) \\
\phi &\sim \mathcal{N}(0, 1) \\
U_{\eta_x} &\sim \mathcal{N}(0, \sigma^2) \in \mathbb{R}^{t_x, 2} \\
U_{\eta_y} &\sim \mathcal{N}(0, \sigma^2) \in \mathbb{R}^{t_y, 2} \\
X &= [\sin(0.5t_x + \phi), \sin(0.5t_x + 2 * \phi] + \eta_x \\
Y &= [\sin(0.5t_y + \phi) + \Delta_0(T, t_y), sin(0.5 * t_y + 2 * \phi) + \Delta_1(T, t_y)] + \eta_y
\end{aligned}
$$

where $\Delta_0(T)$ and $\Delta_1(T)$ are defined as follows:

$$
\Delta_0(T, t) = \begin{cases} \frac{\min(t-20, t_p-20)}{t_p-20} \cdot T & \text{if } U_Z = 0 \text{ or } U_Z = 2 \\ 0 & \text{if } Z = 1 \end{cases}
$$

$$
\Delta_1(T, t) = \begin{cases} \frac{\min(t-20, t_p-20)}{t_p-20} \cdot T & \text{if } U_Z = 1 \text{ or } U_Z = 2 \\ 0 & \text{if } Z = 0 \end{cases}
$$

$t_p = 3$ characterizes the time constant of the treatment response dynamics. The latent categorical variables $U_Z$ represent hidden patient groups that drive the treatment response. In the non-additive noise-response case, we set :

$$
\phi \sim \mathcal{N}(0, \sigma^2)^{40}
$$

such that the phase $\phi$ is time-varying and stochastic. Examples of the trajectories and reconstructed counterfactuals by CFQP are provided in Figure 6.

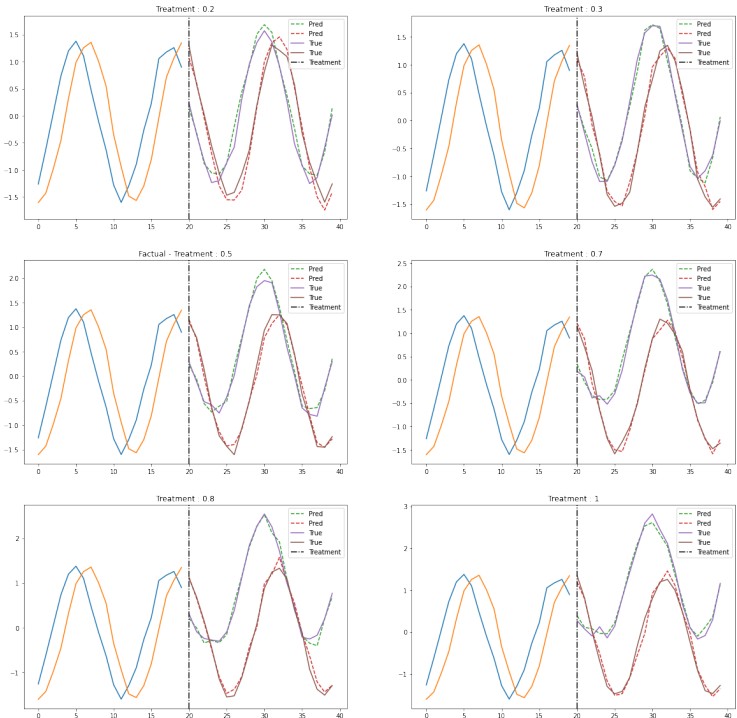

Figure 6: Counterfactuals generated by the model for different levels of treatment. The factual treatment is $T = 0.5$.

### A.3 Cardiovascular Dataset

We use an ODE model of the cardiovascular system as proposed in [29, 6]. Fluid intake is commonly used for treating severe hypotension. However, the response of patient to fluids intake is difficult to assess beforehand. In particular, it depends on the patients cardiac contractility factor and the blood pressure at time of injection. If blood pressure is commonly and easily measured in standard clinical practice, assessing the cardiac contractility level of a patient requires imaging techinques such as echocardiography to measure the stroke volume. Yet, the injection of significant volume of fluids in an irreponsive patients cardiovascular system can lead to severe damage. This lead some clinicians to advocate for fluid challenges, or limited amount of fluid injection to test the responsiveness. This technique is still contested in the medical community and legs raising challenge, much less damaging but also less effective at assessing a patients response has been encouraged. This lack of availability of clear guidelines for fluids intake makes it a perfect case study for counterfactual prediction. Indeed, we'll try to address the question of if a clinician should administer fluids to particular patient based on his clinical history and therefore help informing clinical practice. The system of ODE used to generate the data is the following :

$$\frac{dSV(t)}{dt} = I_{\text{external }(t)}$$

$$\frac{dP_a(t)}{dt} = \frac{1}{C_a}\left(\frac{P_a(t) - P_v(t)}{R_{TPR}(S)} - SV \cdot f_{HR}(S)\right)$$

$$\frac{dP_v(t)}{dt} = \frac{1}{C_v}\left(-C_a\frac{dP_a(t)}{dt} + I_{\text{external }(t)}\right)$$

$$\frac{dS(t)}{dt} = \frac{1}{\tau_{\text{Baro}}}\left(1 - \frac{1}{1 + e^{-k_{\text{width}}\left(P_a(t) - P_{a_{\text{set}}}\right)}} - S\right)$$

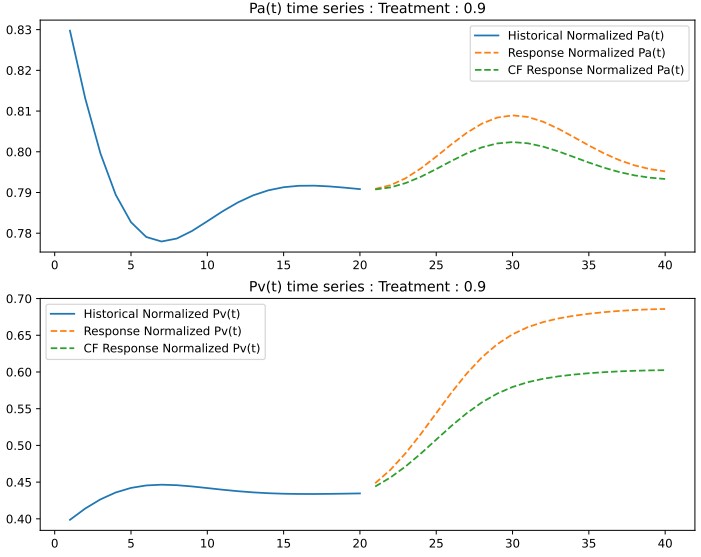

Figure 7: Example of time series in the cardio-vascular data set.

where

$$R_{TPR}(S(t)) = S(t)\left(R_{TPR_{Max}} - R_{TPR_{Min}}\right) + R_{TPR_{Min}} + R_{TPR_{Mod}}$$
$$f_{HR}(S(t)) = S(t)\left(f_{HR_{Max}} - f_{HR_{Min}}\right) + f_{HR_{Min}}.$$

In the above dynamical system, $P_a, P_v, S$ and $SV$ stand for arterial blood pressure, venous blood pressure, autonomic baroreflex tone and cardiac stroke volume respectively. $I_{\text{external }(t)}$ is the amount of fluids given the patient over time and corresponds to the exogeneous input $u_T(t)$ in our model. In the data generation, we model it as

$$I_{\text{external}}(t) = (1 + 2U_Z) \cdot T \cdot 5 \cdot f(P_a(t=0)) \cdot e^{-(\frac{t - t_{\text{treat}} - 5}{5})^2}$$
$$I_{\text{external}}(t) = 0 \quad \forall t \le t_t reat$$

where the treatment assignment $T$ is generated as $T \sim \mathcal{U}(0.6, 1)$ and the hidden group assignment $U_Z \sim \mathcal{B}er(0.5)$ and $t_{\text{treat}}$ is the time of treatment. The function $f$ introduces confounding in the treatment assignment by setting :

$$f(P_a(t=0)) = g(0.5 + (P_a(t=0) - 0.75)/0.1)g(x) \quad = 0.02 \cdot \left((cos)(5x - 0.2) \cdot (5 - x)^2\right)^2$$

We simulate the above system of ODEs for $t_{\text{span}} = 40$ seconds and sample an observation every $\Delta_t = 1$ second. The treatment assignment time is set to $t_{\text{treat}} = 20$.

We then set $X$ as the first two dimensions of the dynamical system $((P_a, P_v)$ for $t < t_{\text{treat}}$ and $Y$ as $((P_a, P_v)$ for $t \ge t_{\text{treat}}$. An example of a generated time series is given in Figure 7.

### A.3.1 Noise Responses

**Additive noise**  In the additive noise case, we add $U_\eta \sim \mathcal{N}(0, \sigma^2)_{\text{span}}^t$ on both $X$ and $Y$

**Non-additive noise**    In the non additive case, we modify the treatment assignment by setting:

$$I_{\text{external}}(t) = (1 + 2U_Z + U_\eta(t)) \cdot T \cdot 5 \cdot f(P_a(t=0)) \cdot e^{-(\frac{t - t_{\text{treat}} - 5}{5})^2}$$

Where $U_{eta}(t)$ is a Gaussian process with zero-mean and diagonal covariance function with variance $\sigma^2$.

# B    Experiments Details

## B.1    Computational Resources

We ran our experiments on cluster containing two types of GPUs : NVIDIA Titan Xp and NVIDIA Quadro GV100. Our experiments resulted in 7 GPU-days of computation.

### B.1.1    Hyper-parameters

In Table 4, we report the hyper-parameters used in our experiments.

| Hyper-parameter | Description | Image | | Harmonic Oscillator | | Cardiovascular | |
|---|---|---|---|---|---|---|---|
| | | additive | non-additive | additive | non-additive | additive | non-additive |
| $lr$ | learning rate | 0.001 | 0.001 | 0.001 | 0.001 | 0.001 | 0.001 |
| $\Delta$ | update period | 10 | 10 | 20 | 20 | 20 | 20 |
| $\sigma$ | noise variance | 0.01 | 0.05 | 0.05 | 0.05 | 0.01 | 0.01 |
| $\text{epochs}_0$ | number of epochs initialization | 50 | 50 | 500 | 500 | 500 | 500 |
| $\text{epochs}_1$ | number of epochs fine-tune | 50 | 50 | 500 | 500 | 500 | 500 |
| $K_0$ | number of classes | 50 | 6 | 6 | 3 | 3 | 2 |
| bs | batch size | 128 | 128 | 128 | 128 | 128 | 128 |
| $N_{\text{train}}$ | Number of train samples | $42,000$ | $42,000$ | 128 | 128 | 500 | 500 |
| $N_{\text{val}}$ | Number of train samples | $9,000$ | $9,000$ | 128 | 128 | 250 | 250 |
| $N_{\text{test}}$ | Number of train samples | $1,000$ | $1,000$ | $1,000$ | 128 | $1,000$ | $1,000$ |

Table 4: Hyper-parameters used for training CFQP on the different datasets.

## B.2    Number of groups

As discussed in section 5.4, the number of groups is an important hyper-parameter in our model. In Table 5,we report the validation reconstruction loss and the corresponding test counterfactual reconstruction MSE for all datasets and different values of $K$. We observe that we obtain a lower validation error for the true value of $K$ in all datasets. In Figures 8a and 8a, we visualize the validation and test performance graphically for the Harmonic Oscillator dataset. We do the same on Figures 9a and 9b for the Cardiovascular dataset and on Figures 10a and 10b for the colored MNIST dataset.

## B.3    Strength of Correlation Experiment

Our approach holds even when the latent background variable $U_Z$ and the covariates $X$ are correlated. When this is the case, information from $X$ can be used to infer the value of $U_Z$. In Figure 4b, we show the impact of the correlation strength $\rho$ between $X$ and $U_Z$.

We incorporate correlation between $X$ and $U_Z$ as follows. The base probability $p_0^k$ of each $U_Z$ is uniform. That is,

$$P^0(U_Z = k) = p_0^k = \frac{1}{K}$$

We modify this probabilty based on the label of the image (the digit label $y_{\text{label}} \in [9]$. For a correlation factor $\rho \in [0, 1]$, we write

| Data | Noise | Number of Centers | Validation MSE | Test MSE |
|------|-------|-------------------|----------------|----------|
| MNIST | Additive | 1 | $0.0171 \pm 0.00012$ | $0.01657 \pm 0.00038$ |
| MNIST | Additive | 2 | $0.02267 \pm 0.00056$ | $0.01094 \pm 0.00025$ |
| MNIST | Additive | 3 | $0.02746 \pm 0.00081$ | $0.00677 \pm 9e-05$ |
| MNIST | Additive | 4 | $0.02552 \pm 0.0022$ | $0.00529 \pm 1e-04$ |
| MNIST | Additive | 5 | $0.02556 \pm 0.01707$ | $0.00377 \pm 0.00016$ |
| MNIST | Additive | 6 | $\mathbf{0.00271 \pm 0.0003}$ | $0.00145 \pm 6e-05$ |
| MNIST | Additive | 7 | $0.00272 \pm 0.00028$ | $0.0016 \pm 7e-05$ |
| MNIST | Additive | 8 | $0.00278 \pm 0.00027$ | $0.00167 \pm 7e-05$ |
| MNIST | Additive | 9 | $0.00289 \pm 0.00031$ | $0.00179 \pm 6e-05$ |
| MNIST | Non-Additive | 1 | $0.01673 \pm 9e-05$ | $0.01673 \pm 0.00026$ |
| MNIST | Non-Additive | 2 | $0.02228 \pm 0.0004$ | $0.01125 \pm 0.00026$ |
| MNIST | Non-Additive | 3 | $0.02632 \pm 0.00063$ | $0.00693 \pm 0.00017$ |
| MNIST | Non-Additive | 4 | $0.03182 \pm 0.00653$ | $0.00552 \pm 0.00017$ |
| MNIST | Non-Additive | 5 | $0.0041 \pm 0.00018$ | $0.004 \pm 0.00018$ |
| MNIST | Non-Additive | 6 | $\mathbf{0.00179 \pm 0.00014}$ | $0.00171 \pm 0.00012$ |
| MNIST | Non-Additive | 7 | $0.00186 \pm 0.00014$ | $0.00185 \pm 0.00011$ |
| MNIST | Non-Additive | 8 | $0.00198 \pm 0.00022$ | $0.00198 \pm 0.00018$ |
| MNIST | Non-Additive | 9 | $0.00212 \pm 0.00023$ | $0.00217 \pm 0.00022$ |
| Harmonic Oscillator | Additive | 1 | $0.1653 \pm 0.01086$ | $0.16584 \pm 0.00443$ |
| Harmonic Oscillator | Additive | 2 | $0.23024 \pm 0.04305$ | $0.08058 \pm 0.01735$ |
| Harmonic Oscillator | Additive | 3 | $\mathbf{9e-05 \pm 3e-05}$ | $0.01332 \pm 0.00123$ |
| Harmonic Oscillator | Additive | 4 | $0.00013 \pm 1e-04$ | $0.0199 \pm 0.00604$ |
| Harmonic Oscillator | Additive | 5 | $0.00035 \pm 0.00024$ | $0.04017 \pm 0.00697$ |
| Harmonic Oscillator | Non-additive | 1 | $0.17451 \pm 0.01053$ | $0.17459 \pm 0.00704$ |
| Harmonic Oscillator | Non-additive | 2 | $0.15164 \pm 0.10242$ | $0.09669 \pm 0.01678$ |
| Harmonic Oscillator | Non-additive | 3 | $\mathbf{0.00891 \pm 0.00035}$ | $0.00956 \pm 0.00096$ |
| Harmonic Oscillator | Non-additive | 4 | $0.00905 \pm 0.00032$ | $0.02279 \pm 0.00327$ |
| Harmonic Oscillator | Non-additive | 5 | $0.00984 \pm 0.00073$ | $0.03464 \pm 0.01158$ |
| CardioVascular | Additive | 1 | $1.01453 \pm 0.06228$ | $1.02569 \pm 0.04794$ |
| CardioVascular | Additive | 2 | $0.03677 \pm 0.01694$ | $0.07701 \pm 0.05001$ |
| CardioVascular | Additive | 3 | $0.04906 \pm 0.02931$ | $0.11363 \pm 0.06759$ |
| CardioVascular | Non-additive | 1 | $1.03865 \pm 0.12068$ | $1.09768 \pm 0.05459$ |
| CardioVascular | Non-additive | 2 | $\mathbf{0.1372 \pm 0.05497}$ | $0.18797 \pm 0.11421$ |
| CardioVascular | Non-additive | 3 | $0.15747 \pm 0.10191$ | $0.2434 \pm 0.13175$ |

Table 5: Reconstruction validation MSE and counterfactual test MSE for all datasets and different number of groups ($K$). Lowest validation MSE are bolded for each dataset.

$$p^k = \begin{cases} \frac{1-((1-p_0)\rho+p_0)}{K-1} & \text{if } k \neq y_{label}\text{mod}(K) \\ ((1-p_0)\rho+p_0) & \text{if } k = y_{label}\text{mod}(K) \end{cases}$$

The above equation suggest that some images with specific label classes will have higher probability of sampling a latent class $U_Z = u_Z$ such that $y_{label}\text{mod}(K) = u_Z$.

In the case when $\rho = 0$, $p^k = p_0^k = \frac{1}{K}$, corresponding to a uniform probability of latent class for each class label. When $\rho = 1$, $p^{y_{label}\text{mod}(K)} = 1$ and the other probabilities are set to 0, corresponding to a deterministic assignment of $U_Z$ conditioned on $y_{label}$.

### B.4 Scalability

CFQPscales linearly with the number of groups $K$, though it can be easily parallelized. Indeed, each sub-model $m_i$ can be evaluated in parallel. Regarding scaling with respect to the dimensions

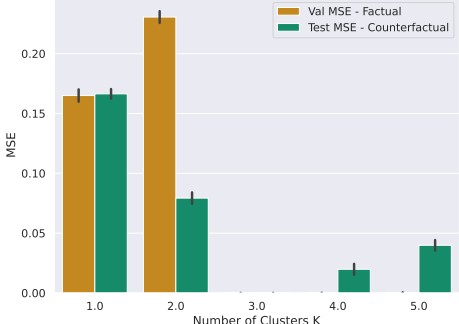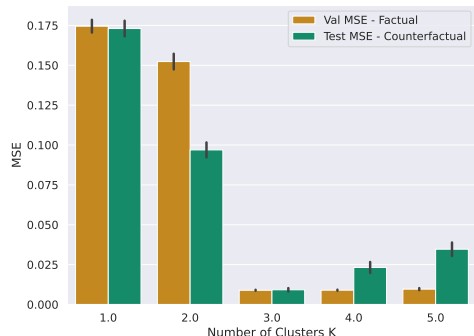

(a) Reconstruction MSE in function of the number of groups in the harmonic oscillator dataset (additive case). The true number of groups is 3.

(b) Reconstruction MSE in function of the number of groups in the harmonic oscillator dataset (non-additive case). The true number of groups is 3.

Figure 8: Analysis of the impact of the number of clusters $K$ on the validation and test performance for the Harmonic Oscillator dataset.

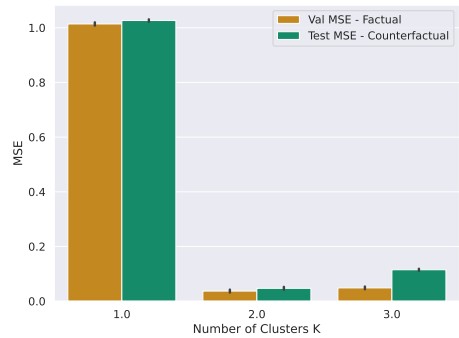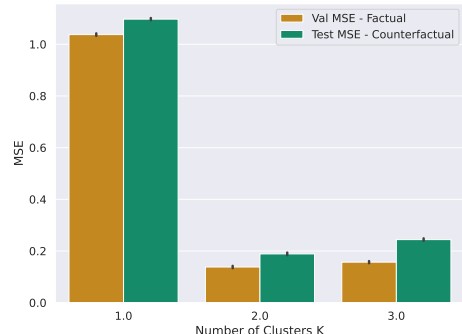

(a) Reconstruction MSE in function of the number of groups in the cardiovascular dataset (additive case). The true number of groups is 2.

(b) Reconstruction MSE in function of the number of groups in the cardiovascular dataset (non-additive case). The true number of groups is 2.

Figure 9: Analysis of the impact of the number of clusters $K$ on the validation and test performance for the Cardiovascular dataset.

of the outcomes and treatments, we can use arbitrary neural networks architectures to process them, for which the complexity can vary. In our experiments, we used CNNs for images, which scale quadratically with the size of the images and TemporalCNNs for time series which scale linearly with the length of the time series as well as number dimensions.

### B.5 Clustering Algorithm

Our approach relies on an *initial* clustering step as described in Section 3.2. In our main experiments, we use a k-means algorithm for this step. However, different clustering strategies can be considered. In Table 6, we report the performance metrics of a variant of our approach using Gaussian Mixture Models (GMM). For computational reasons, the GMM is fit by using a random sub-sample of the training data (1000 samples). We observe only minor performance deviations between k-means and GMM versions of CFQP.

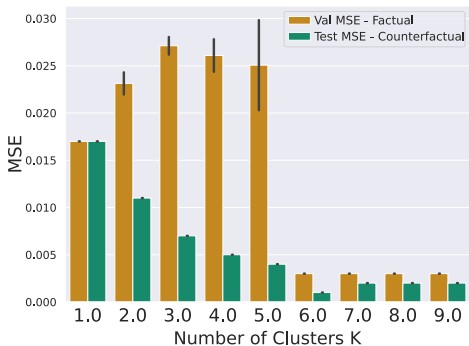 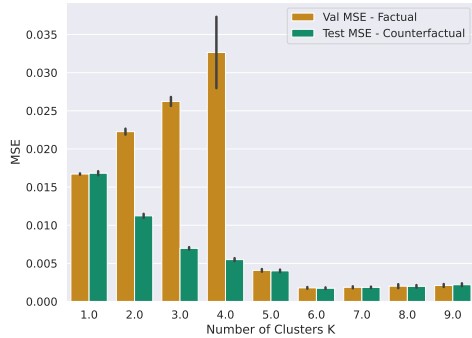

(a) Reconstruction MSE in function of the number of groups in the colored MNIST dataset (additive case). The true number of groups is 6.

(b) Reconstruction MSE in function of the number of groups in the colored MNIST dataset (non-additive case). The true number of groups is 6.

Figure 10: Analysis of the impact of the number of clusters $K$ on the validation and test performance for the colored MNIST dataset.

Table 6: Test MSE of the counterfactual reconstructions for the different datasets.

| | Additive Noise | | | Non-Additive Noise | | |
|---|---|---|---|---|---|---|
| Model | Harmonic Oscillator | colored-MNIST | Cardiovascular | Harmonic Oscillator | colored-MNIST | Cardiovascular |
| Deep-ITE [12] | $0.187 \pm 0.006$ | $0.017 \pm 0.001$ | $1.084 \pm 0.087$ | $0.174 \pm 0.004$ | $0.017 \pm 0.001$ | $1.14 \pm 0.121$ |
| SC [1] | $0.177 \pm 0.131$ | $0.020 \pm 0.001$ | $1.610 \pm 0.141$ | $0.167 \pm 0.004$ | $0.020 \pm 0.001$ | $1.628 \pm 0.144$ |
| Deep-SCM [17] | $0.124 \pm 0.005$ | $0.011 \pm 0.001$ | $0.405 \pm 0.042$ | $0.123 \pm 0.005$ | $0.011 \pm 0.001$ | $0.424 \pm 0.042$ |
| Diff-SCM [24] | $0.082 \pm 0.023$ | $0.008 \pm 0.004$ | $0.206 \pm 0.036$ | $0.106 \pm 0.038$ | $0.009 \pm 0.002$ | $0.311 \pm 0.073$ |
| CFQP (ours) | $0.013 \pm 0.001$ | $\mathbf{0.001 \pm 0.001}$ | $0.077 \pm 0.050$ | $\mathbf{0.009 \pm 0.001}$ | $\mathbf{0.002 \pm 0.001}$ | $\mathbf{0.188 \pm 0.114}$ |
| CFQP-GMM (ours) | $\mathbf{0.001 \pm 0.001}$ | $0.002 \pm 0.001$ | $\mathbf{0.067 \pm 0.042}$ | $\mathbf{0.009 \pm 0.001}$ | $\mathbf{0.002 \pm 0.001}$ | $0.192 \pm 0.0834$ |

# C   Proof of Result 3.1

Our proof is structured as follows. We first show identifiability of non-parameteric mixtures at a single point $((X = x, T = t)$. We then show identifiability of the permutation function over $(\mathcal{X} \times \mathcal{T})$. Finally, based on these results, we derive the bound of Result 3.1.

## C.1   Identifiability of Non-Parametric Mixture Models.

To show the identifiability in the non-additive case, we re-use notation and start from the outline derived in Aragam et al. [3].

### C.1.1   Identifiability Result

We let $(X, d)$ be a metric space and $(\mathcal{P}(X), \rho)$ the space of regular Borel probability measures on $X$ with finite $r$th moments metrized by a metric $\rho$. Let's further define $\mathcal{P}^2(X) = \mathcal{P}(\mathcal{P}(X))$, the space of mixing measures over $\mathcal{P}(X)$. We represent a mixture distribution over $X$ as a new probability measure $m(\cdot, \Lambda) \in \mathcal{P}(X)$ such that $m(A; \Lambda) = \int \gamma(A) d\Lambda(\gamma)$ for any $A \subset X$.

Given a Borel set $\mathfrak{L} \subset \mathcal{P}^2(X)$, we define $\mathcal{M}(\mathfrak{L}) := \{m(\Lambda) : \Lambda \in \mathfrak{L}\}$ as a family of mixture distributions over $X$. Because we are interested in finite mixtures, we use $\mathcal{P}_k^2 := \{\Lambda \in \mathcal{P}^2(X) : \|\text{supp}(\Lambda)\| \leq k\}$ to denote the set of mixing measures with $k$ components. Finite mixtures $\Gamma \in \mathcal{M}(\mathcal{P}_k^2(X))$ can thus also be written in a more intuitive form, *i.e.* $\Gamma = \sum_{k=1}^K \omega_k \gamma_k$ where $\gamma_k \in \mathcal{P}(X)$ are the *mixture components* and $\omega_k$ are mixing weights, as in Equation 2 in the main text.

Note that we did not make any assumption regarding the distribution of the mixture components $(\gamma_k \in \mathcal{P}(X))$. Now, from [3], we have the following result.

**Theorem C.1.** *If $\mathfrak{L}$ is a $\mathfrak{Q}_L$-clusterable family, then there exists a function $h : \mathcal{M}(\mathfrak{L}) \to \mathfrak{L}$ such that $h(m(\Lambda)) = \Lambda$, where $m : \mathfrak{L} \to \mathcal{M}(\mathfrak{L})$ is the canonical embedding. In particular, $m$ is a bijection and the mixture model $\mathcal{M}(\mathfrak{L})$ is identifiable.*

We see that identifiability of the mixture therefore hinges on the notion of *clusterability* of the individual distributions that compose the mixture. We now make the notion of clusterability more precise.

### C.1.2 Clusterability

**Projections**  Let's first introduce $\{\mathfrak{Q}_L\}_{L=1}^{\infty}$ as an indexed collection of families of mixing measures that satisfy the following requirements:

- $\mathfrak{Q}_L \subset \mathcal{P}_L^2(X)$ for each $L$;
- $\{\mathfrak{Q}_L\}$ is a filtration ($\mathfrak{Q}_L \subset \mathfrak{Q}_{L+1}$);
- The collection of mixture distributions $\mathcal{M}(\mathfrak{Q}_L)$ is identifiable for each $L$.

Examples of such a collection is the filtration of Gaussian mixtures with $L$ components.

We then proceed by defining the $\rho$-projection of a mixture distribution $\Gamma = m(\Lambda)$ on a family of mixture distribution $\mathfrak{Q}_L$ by

$$T_L\Gamma = \{Q \in \mathfrak{Q}_L : \rho(Q,\Gamma) \leq \rho(P,\Gamma) \forall P \in \mathfrak{Q}_{\mathcal{L}}\}$$

where $\rho$ is a metric on $\mathcal{P}(X)$ (*e.g.* the Hellinger distance). This projection intuitively maps a mixture distribution $\Gamma$ to the closest element of the family $\mathfrak{Q}_L$ it is projected upon. We futher define the map $M_L : \mathfrak{Q}_{\mathcal{L}} \to \mathfrak{Q}_L$ that a projected distribution to its mixing measure:

$$M_L\Gamma = M_L(T_L\Gamma) = \{\Omega \in \mathfrak{Q}_L : m(\Omega) \in T_L\Gamma\}$$

**Assignment Functions**  Because it is an element of $\mathfrak{Q}_L$, the projection $T_L\Gamma = Q^*$ is a mixture distribution and can thus be written as $T_L\Gamma = \sum_{l=1}^{L} \omega_l^* \gamma_l^*$. When projecting on mixture distributions with more mixture components than in the original distribution, *i.e.* $L \geq K$, one can then assign mixture components in $Q^*$ to mixture components in $\Gamma$. We define the set of assignment maps $\alpha : [L] \to [K]$ as $\mathbb{A}_{L \to K}$. Given such an assignment map, we define $\psi_k(\alpha) := \sum_{l \in \alpha^{-1}(k)} \omega_l$ for the weights induced by the assignment map and $Q_k(\alpha) := \frac{1}{\psi_k(\alpha)} \sum_{l \in \alpha^{-1}(k)} \omega_l^* \gamma_l^*$ for the induced mixing components.

**Regularity of a mixing measure**  We are now ready for introducing the definition of regularity of a mixing measure.

**Definition C.1** (Regularity). Suppose $\Lambda \in \mathcal{P}_K^2(X)$ and $\Gamma = m(\Lambda) \in \mathcal{M}_K(\mathcal{P}^2(X))$. The mixing measure $\Lambda$ is called $\mathfrak{Q}_L$-regular if:

- (a) The $\rho$-projection $Q^* = T_L\Gamma$ exists and is unique for each $L$ and $\lim_{L \to \infty} T_L\Gamma = \Gamma$;

- (b) There exists a sequence of assignment functions $\alpha = \alpha_L \in \mathbb{A}_{L \to K}$ such that $\lim_{L \to \infty} Q_k^*(\alpha) = \gamma_k$ and $\lim_{L \to \infty} \psi_k^*(\alpha) = \omega_k \quad \forall k = 1, \dots, K$

Any assignment function where (b) holds is called a *regular assignment*.

**Clusterability**  We can now finally define clusterability of a family of mixing measures with respect to another regular family of mixing measures.

**Definition C.2** (Clusterable family). A family of mixing measures $\mathfrak{L} \subset \mathcal{P}^2(X)$ is called a $\mathfrak{Q}_L$-clusterable family, or just a clusterable family, if

- (a) $\Lambda$ is $\mathfrak{Q}_L$-regular for all $\Lambda \in \mathfrak{L}$;

- (b) For all sufficiently large $L$, there exists a function $\chi_L : M_L(\mathfrak{L}) \to \mathbb{A}_{L \to K}$ such that $\chi_L(\Omega^*)$ is a regular assignment for every $\Lambda \in \mathfrak{L}$, with $\Omega^* = M_L(T_L \Gamma)$.

The resulting mixture model $\mathcal{M}(\mathfrak{L})$ is called a clusterable mixture model.

More intuitively, a family of mixing measures will then be $\mathfrak{Q}_L$-clusterable if one can project every mixing measure of the family onto $\mathfrak{Q}_L$ and if, for each of these projections, there exists a function that outputs a regular assignment for each mixture distribution given as input. This regular assignment is such that it clusters the elements of $Q^*$ in a way that the distribution of each cluster converges to the distribution of a element of the original mixture distribution. In this case, one can *identify* a family of mixing measures by projecting each mixing distribution onto a regular family (*e.g.* Gaussian mixtures) and cluster it accordingly to $\chi_L$. We refer the interested reader to [3] for more insight on clusterability and identifiability in non-parametric mixtures.

## C.2 Proof of result 3.1

### C.2.1 Identifiability at a Fixed Point

We start our proof by recalling the identifiability result in the case of clusterable families of mixing measures, whose proof can be found in [3].

**Theorem C.2.** *If $\mathfrak{L}$ is a $\mathfrak{Q}_L$-clusterable family, then there exists a function $h : \mathcal{M}(\mathfrak{L}) \to \mathfrak{L}$ such that $h(m(\Lambda)) = \Lambda$, where $m : \mathfrak{L} \to \mathcal{M}(\mathfrak{L})$ is the canonical embedding. In particular, $m$ is a bijection and the mixture model $\mathcal{M}(\mathfrak{L})$ is identifiable.*

As laid out in the main text, we assume three observed random variables $X \in \mathcal{X}$, $Y \in \mathcal{Y}$ and $T \in \mathcal{T}$. As suggested by Assumption 1 in the main text, the conditional distribution of $Y$ at each point $(X = x, T = t)$ is given by an unknown mixture distribution from a family $\mathcal{M}_K(\mathfrak{L})$ where each mixing measure $\Lambda \in \mathfrak{L}$ satisfies $\Lambda \in \mathcal{P}^2(Y)$ and $|\operatorname{supp}(\Lambda)| = K$. We write the distribution of $Y$ conditioned on $X$ and $T$ as:

$$Y \mid X, T \sim \gamma(X, T) = \sum_{k=1}^{K} \omega_k \gamma_k(X, T) \quad \text{with } \gamma(X, T) \in \mathfrak{L} \tag{6}$$

Assuming that $\mathfrak{L}$ is clusterable, we can use Theorem C.2 to deduce that the mixture distribution $\gamma(X, T)$ is identifiable at each fixed point $(X = x, T = t)$.

Importantly, the idenfitifiablity insured by Theorem C.2 is up to a permutation of the mixture components. We write $\sigma_{x,t} : [K] \to [K]$ the permutation function at a specific point $(X = x, T = t)$. We define the assignment function that maps the elements of a mixture distribution $\gamma(X, T)$ to $U_Z$ as $F_{(x,t)} : \mathcal{M}(\mathfrak{L}) \to \{[K], \mathcal{P}(Y)\}^K$. We now move to showing that the assignment function is identifiable up to a constant permutation $\bar{\sigma}$.

### C.2.2 Identifiability of the Assignment Function

The support of random variables $X$ and $T$, $\mathcal{X} \times \mathcal{T}$ is connected. Assuming further (Assumption 3) that $\mathfrak{L}$ is a clusterable family, we know that there exists a cluster function $\chi_{L,(x,t)} : \mathcal{M}_{\mathcal{L}}(\mathfrak{L}) \to \mathbb{A}_{L \to K}$ that maps components of the projected mixture onto the initial one. We can then define a combined operator that take a mixture distribution as input and returns an indexed sequence of mixture components.

Let $F_{(x,t)} : \mathcal{M}(\mathfrak{L}) \to \{[K], \mathcal{P}(Y)\}^K$ be this operator consisting of applying a $\rho$-projection onto a regular family, cluster the result according to $\chi_{L,(x,t)}$ and return the indexed estimated distributions $Q_k^*(\alpha)$. As the per the definition of regular mixing measures, we have also have $\lim_{L \to \infty} Q_k^*(\alpha) = \gamma_k$. Note that the indexing of the elements of the mixing distribution is arbitrary at a single point $(X = x, T = t)$ and thus defined up to a permutation $\bar{\sigma}$.

Because $X$ and $T$ are continuous on a connected domain, we can evaluate the operator at $(X = x + \delta x, T = t + \delta_t)$.

$$F_{(x,t)} = \{(\sigma_{x,y}(i), q_i^*(x,t)) : i = 1, ..., K\}$$
$$F_{(x+\delta x, t+\delta t)} = \{(\sigma_{x+\delta x, y+\delta y}(i), q_i^*(x + \delta x, t + \delta t)) : i = 1, ..., K\}$$

What is more, for arbitrary small $(\delta x, \delta t)$ and $\forall (x,t) \in (\mathcal{X} \times \mathcal{T}); \forall k, k', k'' \in [K]$, we have

$$\rho(\gamma_k(x,t), \gamma_{k'}(x,t)) > \rho(\gamma_{k''}(x,t), \gamma_{k''}(x + \delta x, t + \delta t)$$

because we require the moments of each $\gamma_k$ to be continuous in $(x,t)$. It therefore implies a relation between $\sigma_{x,y}$ and $\sigma_{x+\delta x, y+\delta y}$ that satisfies:

$$\sigma_{x+\delta x, y+\delta y}(i) = \text{argmin}_j \rho(q_i^*(x,t), q_j^*(x + \delta x, t + \delta t))$$

The assignment function $F_{(x,t)}$ is thus identifiable up to a constant permutation $\bar{\sigma}$ that will determine all other permutations in $\mathcal{X} \times \mathcal{T}$.

### C.2.3 Counterfactual Identifiability

To recapitulate, in the previous sections, we have shown (1) identifiability of a non-parametric mixture at a point $(X = x, T = t)$ based on the Assumption 3 and (2) identifiability of the permutation function over $(\mathcal{X} \times \mathcal{T})$. That is, for each point in $(X = x, T, t)$, we can identify the mixture components and the mixture weights and associate it to any other point $(X = x', T =')$, through the assignment function $F_{(x,t)}$.

In particular, we can identify the means $\mu_k$ of each cluster component $\gamma_k$. We have

$$\lim_{N \to \infty} \hat{\mu}_k(X, T) = \mathbb{E}_{Y \sim \gamma_k(X,T)}[Y] = \mu_k(X, T) \tag{7}$$

and

$$\lim_{N \to \infty} \hat{\omega}_k(X, T) = \omega_k(X, T). \tag{8}$$

Because the above estimators converge, the following posterior probability converges to the true value as well:

$$P(U_Z = u_Z \mid Y = y, X = x, T = t) \tag{9}$$
$$= \frac{P(Y = y \mid U_Z = u_Z, X = x, T = t)P(U_Z = u_Z, \mid X = x, T = t)}{\sum_{k=1}^K P(Y = y \mid U_Z = k, X = x, T = t)P(U_Z = k, \mid X = x, T = t)} \tag{10}$$

for all $x \in \mathcal{X}$, $t \in \mathcal{T}$ and $y \in \mathcal{Y}$. Indeed, $P(Y = y \mid U_Z = u_Z, X = x, T = t)$ is identifiable, and $P(U_Z = u_Z, \mid X = x, T = t) = \omega_{u_Z}(x, t)$.

Let $\hat{\omega}_{u_Z}(x, y, t)$ be the estimator of $P(U_Z = u_Z \mid Y = y, X = x, T = t)$ We are now ready to define the estimator for the counterfactual distribution $\nu_{t'}(x, y, t)$.

$$\nu_{t'}(x, y, t) = \sum_k \hat{\omega}_k(x, y, t)\delta(y = \hat{\mu}_k(x, t')). \tag{11}$$

This estimator is a discrete mixture distribution with mass on the means of the different mixture components, weighted by the posterior probability of the initial sample $(x, t, y)$ belonging to a particular component.

### C.2.4 Bounds

We can now use this estimator $\nu_{t'}(x, y, t)$ to derive the bounds of Result 3.1. We first write the $W_1$ distance between the estimated and true counterfactual distributions:

$$W_1(\nu_{t'}(x, t, y), Y'(x, t, y)) \tag{12}$$

with

$$\nu'_t(x, t, y) = \sum_{k=1}^{N} P(U_Z = k \mid X = x, T = t, Y = y)\delta(\nu_{t'} = \hat{\mu}_k(X = x, T = t')) \tag{13}$$

$$Y'(x, t, y) = \sum_{k=1}^{K} P(U_Z = k \mid X = x, T = t, Y = y) \tag{14}$$

$$\cdot P(Y' = y' \mid X = x, T = t, Y = y, U_Z = k) \tag{15}$$

We thus need to compute the $W_1$ distance between two mixture distributions where each component is scaled by the same factor. By restricting the transport map to assigning each mixture component $k$ of $\nu'_t$ to the respective one in $Y'$, we have

$$W_1(\nu_{t'}(x, t, y), Y'(x, t, y)) \tag{16}$$

$$\leq \sum_{k=1}^{K} P(U_Z = k \mid x, t, y)W_1(\delta(\nu_{t'} = \hat{\mu}_k(x, T = t')), P(Y' = y' \mid x, t, y, U_Z = k)). \tag{17}$$

Now, we can write the individual components of the above sum as follows

$$
\begin{aligned}
&P(U_Z = k \mid x, t, y)W_1(\delta(\nu_{t'} = \hat{\mu}_k(x, T = t')), P(Y' = y' \mid x, t, y, U_Z = k)) \\
&\leq P(U_Z = k \mid x, t, y) \\
&\quad \cdot \iint_{y',\nu_{t'}} \|y' - \nu_{t'}(x, t, y)\|_2 \delta(\nu_{t'} = \hat{\mu}_k(x, T = t'))P(Y' = y' \mid x, t, y, U_Z = k))\, dy'\, d\nu_{t'} \\
&= P(U_Z = k \mid x, t, y) \int_{y'} \|y' - \hat{\mu}_k(x, T = t')\|_2 P(Y' = y' \mid x, t, y, U_Z = k))\, dy' \\
&= \int_{y'} \|y' - \hat{\mu}_k(x, T = t')\|_2 P(Y' = y' \mid x, t, y, U_Z = k)) \\
&\quad \cdot \frac{P(Y \mid x, t, U_Z = k)P(U_Z = k \mid x, t)}{P(Y \mid x, t)}\, dy'
\end{aligned}
$$

where we bounded the $W_1$ distance by using the joint probability distribution as the transport map between $\nu'_t$ and $Y'$.

Because the $W_1$ distances are positive, the inequality applies to the expectation and we can write

$$
\begin{aligned}
\mathbb{E}_Y[W_1(\nu_{t'}(x, t, Y), Y'(x, t, Y))] &\leq \\
\sum_{k=1}^{K} \iint_{y',y} \|y' - \hat{\mu}_k(x, T = t')\|_2 &P(Y' = y' \mid x, t, Y, U_Z = k))\cdot \\
\frac{P(Y \mid x, t, U_Z = k)P(U_Z = k \mid x, t)}{P(Y \mid x, t)} &P(Y \mid x, t)\, dy'\, dy
\end{aligned}
$$

We finally bound the above expectation by marginalizing $Y$:

$$\mathbb{E}_Y[W_1(\nu_{t'}(x,t,Y), Y'(x,t,Y))] \leq \tag{18}$$

$$\sum_{k=1}^K \iint_{y',y} \|y' - \hat{\mu}_k(x, T = t')\|_2 P(Y' = y' \mid x, t, Y, U_Z = k) \cdot$$

$$P(Y \mid x, t, U_Z = k) P(U_Z = k \mid x, t) \, dy' \, dy \tag{19}$$

$$= \sum_{k=1}^K P(U_Z = k \mid x, t) \int_{y'} \|y' - \hat{\mu}_k(x, T = t')\|_2 P(Y' = y' \mid x, t, U_Z = k) \, dy' \tag{20}$$

$$= \sum_{k=1}^K P(U_Z = k \mid x, t) \int_{y'} \|y' - \hat{\mu}_k(x, T = t')\|_2 \gamma_k(y' \mid x, t, U_Z = u_Z) \, dy' \tag{21}$$

$$= \sum_{k=1}^K P(U_Z = k \mid x, t) W_1^k(\hat{\mu}_k, \gamma_k) \tag{22}$$

$$\leq \max_k W_1^k(\hat{\mu}_k, \gamma_k) \tag{23}$$

$$= \delta$$

Where 23 holds because it 22 is a convex combination $W_1^k(\hat{\mu}_k, \gamma_k)$. This ends the proof of Result 3.1.

### C.2.5 The additive noise case

Result 3.1 mentions the bounds reduces to zero $\delta = 0$ when the noise is additive. From identifiability at of a mixture at a single point, we had:

$$\lim_{N \to \infty} \hat{\mu}_k(x,t) = \mathbb{E}_{Y \sim \gamma_k(x,t)}[Y] = \mu_k(x,t), \tag{24}$$

$$\lim_{N \to \infty} \hat{\omega}_k(x,t,y) = P(U_Z = u_Z \mid x, t, y). \tag{25}$$

We also have convergence of the covariance matrix estimator

$$\lim_{N \to \infty} \hat{\Sigma}_k(x,t) = \mathbb{V}\mathrm{ar}_{Y \sim \gamma_k(x,t)}[Y] = \Sigma_k(x,t). \tag{26}$$

In the additive noise setup, the distribution of $Y$ being normally distributed, the value of $U_\eta$ is actually identifiable. Let us recall the definition of the counterfactual distribution (Eq. 1) in the main text.

$$P(Y_{t'} = y' \mid X = x, Y = y, T = t) = \int_u P(Y_{t'}(u) = y') P(U = u \mid X = x, T = t, Y = y)$$

In the additive case, we have $Y \mid X = x, T = t, U_Z = u_Z = \Sigma_{u_Z}(x,t) U_\eta + \mu_k(x,t)$ with $U_\eta \sim \mathcal{N}(0, \mathbf{1})$. The counterfactual density can thus be written as

$$\nu_{t'}(x,t,y) = \sum_k P(U_Z = k \mid x, y, t) \cdot \delta(u_{\eta,k} = \Sigma_k(x,t)^{-1} y - \mu_k(x,t))$$

$$\cdot \delta(y' = \Sigma_k(x,t') u_{\eta,k} + \mu_k(x,t')) \tag{27}$$

Considering the following estimator for the counterfactual distribution:

$$\hat{\nu}_{t'}(x,t,y) = \sum_k \hat{\omega}_k(x,t,y) \cdot \delta(u_{\eta,k} = \hat{\Sigma}_k(x,t)^{-1}y - \hat{\mu}_k(x,t)) \cdot \delta(y' = \hat{\Sigma}_k(x,t')u_{\eta,k} + \hat{\mu}_k(x,t'))$$

We can use Equations 24, 25 and 26 to show that our estimator converges to the true distribution in the limit of infinite number of samples:

$$\lim_{N \to \infty} \hat{\nu}_{t'}(x,t,y) = \nu_{t'}(x,t,y) \tag{28}$$

Therefore, the distance between the estimated and true counterfactual distributions reduces to 0:

$$\lim_{N \to \infty} W_1(\hat{\nu}_{t'}(x,t,y), \nu_{t'}(x,t,y)) = 0. \tag{29}$$

This concludes our proof of Result 3.1.