# OpenReview forum: "Deep Counterfactual Estimation with Categorical Background Variables"
_NeurIPS.cc/2022/Conference — NeurIPS 2022 Accept_

### Official Review · Reviewer_PhWG · 2022-06-16

**Rating:** 6
**Confidence:** 3
**Soundness:** 2 fair
**Presentation:** 3 good
**Contribution:** 3 good

**Summary:**

The paper proposes a series of assumptions that permit identifiability in cases with categorical background variables. It further proposes CFQP a neural net based algorithm for counterfactual estimation based on the aforementioned assumptions and an EM method

**Questions:**

- What is the justification of gaussian assumptions and not something like $g(u^{*}_{\eta})$ where is a learnable function and u* an approximation
- What is the approximate complexity of the proposed NN method? How does the it scale as the number of treatments, outcomes and categories scale ?
- What is the accuracy of the method in the cases where the ground truth is known ? in the image scenario what is the SSIM ?

**Limitations:**

The authors have explicitly stated their assumptions and provided an ethical discussion

**Strengths And Weaknesses:**

- Originality : the paper appears to provide a novel and original methodology both in terms of identifiability results and NN methodology
- Clarity : adequate clarity
- Significance: Potential for impact in the wider causal ML community
- Related works : Not enough discussion of related works and how this work relates to it
- Experimentation: Only MSE and some images are shown - no accuracy for counterfactuals where ground truth exists - synthetic ones
- Methodology: good but assumption of Gaussian distributions is not supported. See questions for more

Disclaimer: I did not check the validity of the proofs

---

> ### Author Response · Authors · 2022-07-28
> **Answer to reviewer's comments**
>
> Dear Reviewer PhWG
>
> We thank you for your encouraging and pertinent review. We hope the following will be answer your questions. In any case, please let us know if you have any other enquiry about our work. We are also happy to include other works that you would find relevant to our paper.
>
> ### About reporting accuracy for the counterfactuals.
> > Only MSE and some images are shown - no accuracy for counterfactuals where ground truth exists - synthetic ones
>
> **Answer** As we mention in Section 5.1 of the paper, counterfactual evaluation is challenging, because counterfactuals are, by definition, not observed. We thus follow previous works here in evaluating our method on synthetic datasets (like [Pawlowski2020], which is the inspiration for the image experiment), for which we have ground truth. Accuracy cannot be reported because we evaluated counterfactual reconstruction of (potentially continuous) high-dimensional objects (like time series or images). We thus believe that MSE (or PEHE) provides a meaningful evaluation of the counterfactual reconstruction ability.
>
> ### About the Gaussian noise assumption
> > What is the justification of gaussian assumptions and not something like $g(u_{\eta})$ where is a learnable function and $u$ an approximation
>
> **Answer** The assumption of gaussian noise variables $U_{\eta}$ is meant to be both absolutely general and practical. General because we indeed consider that the noise Un can be processed by an arbitrary function g as you suggested. In our paper, we use the notation $f_Y(x,t,u_z,u_{\eta})$ (lines 117-118). This potentially non-linear function can thus model the gaussian variables into an arbitrary distribution.
>
> In the special additive case, on the other hand, we assume the *output* variables are normally distributed. This special case of our general framework enjoys favorable theoretical properties (such as exact identifiability of the counterfactual distribution), so we thought it would be meaningful to discuss this special case. Nevertheless, our results and experiments hold in the general case, where the output distribution is arbitrary.
>
>
> ### About the scalability of our method
> > What is the approximate complexity of the proposed NN method? How does the it scale as the number of treatments, outcomes and categories scale ?
>
> **Answer** It scales linearly with the number of categories, though it can be easily parallelized. Indeed, each sub-model $m_i$ can be evaluated in parallel.
>
> Regarding scaling with respect to the dimensions of the outcomes and treatments, we can use arbitrary neural networks architectures to process them, for which the complexity can vary. In our experiments, we used CNNs for images, which scale quadratically with the size of the images and TemporalCNNs for time series which scale linearly with the length of the time series as well as number dimensions.
>
> We added a paragraph on scalability in Appendix section B.4.
>
> ### About structural similarity index measure
> >  in the image scenario what is the SSIM ?
> -  We welcome your suggestion of evaluating the structural similarity index in the images setup, which can give a better measure of the quality of the image reconstruction.  We therefore provide evaluation of the SSIM in the appendix (Table 3 in Section A.1.1). We report the numbers here as well for convenience:
>
> | Model | Additive noise  | Non-additive noise  |
> |---|---|---|
> | Deep-ITE |  $0.490 \pm 0.006$| $0.484 \pm 0.004$ |
> | Synthetic Controls | $0.020 \pm 0.001$ | $0.020\pm0.001$ |
> | Deep-SCM | $0.857\pm0.003$ | $0.856\pm0.004$  |
> | Diff-SCM | $0.780\pm0.063$ | $0.750\pm0.052$  |
> | CFQP(ours)| $0.964\pm0.003$ | $0.959\pm0.005$  |
>
> We observe that our method clearly outperforms all other baselines in this metric as well.

---

### Official Review · Reviewer_Ddey · 2022-07-11

**Rating:** 7
**Confidence:** 4
**Soundness:** 3 good
**Presentation:** 3 good
**Contribution:** 4 excellent

**Summary:**

The paper deals with the problem of counterfactual identification and estimation in treatment effect estimation problems from observational datasets. The task of counterfactual identification is challenging as we need access to the structural equations, which is a strong assumption in typical real-world datasets. The authors tackle these issues by making the assumption of discrete/categorical latent variables that affect the final outcome distribution and provide novel identification results under the ($\rho$, $\delta$) definition. They propose the approach CFQP, which takes inspiration from the discrete latent assumption and trains a mixture model using expectation maximization. They show the gains of the proposed approach against the existing works in the literature on semi-synthetic datasets; MNIST, harmonic oscillator, and cardiovascular simulator.

**Questions:**

-  How does the proposed approach perform when there is a mismatch in the total number of latent categories for the harmonic oscillator and the cardiovascular dataset? The authors only show the effect of error in modeling the total latent categories for the MNIST dataset. Further, the authors should report results for the approach CFQP when they tune the total latent categories ($\hat{K}$) using validation data (instead of selecting $\hat{K}=K$) and include the same across different settings in Table 1.

- A standard way to define counterfactual identification (https://causalai.net/r60.pdf) would be as follows: 'if two different SCM $M_1, M_2$ have the same causal graph and generate same observational/interventional distributions, then a counterfactual query Q is identifiable iff $M_1(Q) = M_2(Q)$'. How does the ($\rho$, $\delta$) definition considered by the authors compare with it? Does the identification result in the paper imply there can still be some SCMs that are not distinguishable from each other considering the proposed assumptions, but they yield different results on the counterfactual query?

**Limitations:**

I think the authors have adequately addressed the limitation and potential negative societal impact of their work.

**Strengths And Weaknesses:**

**Strengths**

- Given that we would never observe structural causal models (SCM) for a majority of realistic datasets, the task of counterfactual identification without access to SCM is an extremely relevant problem. The paper provides identification results towards the same under the discrete latent variable assumption, which is novel to the best of my knowledge.

- The identification results are significant as the assumptions introduced in the paper for counterfactual identification are practically motivated. Many practical scenarios can have discrete latent variables affecting the outcome distribution, and additionally, there are no restrictions in terms of the dimensionality of treatment and outcome.

- The proposed approach (CFQP) has simplicity in the sense of being inspired from mixture models, and obtains substantial improvements over the prior approaches.

- The paper is overall well written with good explanations for all the assumptions and nice presentations of the empirical results.

**Weaknesses**

- The proposed approach (CFQP) required the knowledge of the true number of latent categories (K). While the authors present ablation studies when there is a mismatch with the true total latent categories, further empirical analysis might be needed.

- The identification result holds for the ($\rho$, $\delta$) definition stated in the paper, which might not lead to perfect identification.

Please refer to the question section below for further details.

Note: I do think the paper makes good contributions and I am happy to raise my score if the authors answer my concerns with the evaluation studies.

---

> ### Author Response · Authors · 2022-07-29
> **Answers to reviewer's comments**
>
> Dear Reviewer Ddey,
>
> Thank you for your encouraging review and for the great suggestions that help us strengthen our paper. We are glad to report that we have implemented all your suggestions in the updated version of the paper. Please find more details below.
>
> ### About consistently using K as an hyper-parameter.
> > How does the proposed approach perform when there is a mismatch in the total number of latent categories for the harmonic oscillator and the cardiovascular dataset? The authors only show the effect of error in modeling the total latent categories for the MNIST dataset. Further, the authors should report results for the approach CFQP when they tune the total latent categories ($\hat{K}$) using validation data (instead of selecting $\hat{K}=K$) and include the same across different settings in Table 1.
>
> - This is a great point and we are happy to report a stronger experimental design in the updated version of the paper. We re-ran the model with multiple values of $K$ and now report the results with the value of $K$ that resulted in the lowest validation error. It appears that validation error was lowest for the true value of $K$ in all experiments. Intuitively, this can be explained by the fact that a lower value results in not enough expressivity for modeling the data well, while an higher value of K will result in a lower number of data points per sub-model $m_i$, therefore hampering performance. For sake of completeness, we have included the validation and test counterfactual MSE of all models for the different values of $K$ in section B.2. along with the corresponding graphs (Figures 8, 9 and 10). On these figures, we can clearly see a sharp drop in MSE at the exact value of $K$.  We hope this addresses your concerns regarding our evaluation pipeline and that it will have convinced you of the pertinence of our approach.
>
> ### About the connection with other definitions of counterfactual identification
> > A standard way to define counterfactual identification (https://causalai.net/r60.pdf) would be as follows: 'if two different SCM $M_1,M_2$ have the same causal graph and generate same observational/interventional distributions, then a counterfactual query $Q$ is identifiable iff $M_1(Q)=M_2(Q)$. How does the $\rho,\sigma$ definition considered by the authors compare with it? Does the identification result in the paper imply there can still be some SCMs that are not distinguishable from each other considering the proposed assumptions, but they yield different results on the counterfactual query?
>
> - Rather than asking for exact identifiability, our approach attempts to quantify the discrepancy between the different counterfactual distributions. That is, we relax the identifiability requirements by bounding the expected distance between the different counterfactual distributions. Instead of asking for perfect counterfactual reconstruction, we ask how much it is expected to deviate from the truth. This motivates our $(\rho,\delta)$ definition as it allows to introduce a bound on the distance $\delta$ between the true and estimated distributions. *We believe that exact identifiability or reconstruction is not always required and that particular applications allow for approximate reconstruction, up to some bound on the error*.
>
>     This relaxation further allows to avoid have more general functional form for the structural equations (e.g. modeled by neural network) at the expense of some controlled error in the counterfactual reconstruction.
>
>     Our result is more easily compared with your suggested definition of identifiability in the special case of additive noise that we cover in the paper. In this case our results show that the counterfactuals are identifiable with distance 0, which then coincides with the above definition. This can be seen in the following way. Because we don’t observe the counterfactuals (by definition), our model is trained on the observed data points and thus maximize the likelihood of the observational distribution. Let M1 and M2 be two such models that are trained on the data. Because our estimator converge to the true counterfactual distribution, it results that M1(q) = M2(q) for any counterfactual query. Hence, they are identifiable in the sense you mentioned.

---

> ### Author Response · Authors · 2022-08-09
> **Did we address all your concerns ?**
>
> Dear Reviewer Ddey,
>
> Thanks again for taking the time to review our paper and for your encouraging feedback ! As the discussion period is coming to a close, may we enquire if all the concerns you raised have been properly addressed ? Thank you very much,
>
> Best Regards,
>
> The authors

---

> > ### Comment · Reviewer_Ddey · 2022-08-09
> > **Good response**
> >
> > Thanks for the well-written response, it addresses the concerns I had.

---

> > > ### Author Response · Authors · 2022-08-09
> > > **Thank you very much !**
> > >
> > > Thank you so much for your feedback and for updating your score ! It's truly appreciated ! We are glad our response addressed your concerns.
> > >
> > > Best Regards,
> > >
> > > The authors

---

### Official Review · Reviewer_dSw5 · 2022-07-11

**Rating:** 4
**Confidence:** 5
**Soundness:** 2 fair
**Presentation:** 3 good
**Contribution:** 2 fair

**Summary:**

The authors tackle the limitation of pre-existing neural network-based structual equation methods which only enable to model the factual distribution, not counterfactual, such that it often leads to incomplete ounterfactual distributions. The authors focus on the counterfactual inference of treatment effects on patient trjectories in healthcare setting. They validate the proposed model on three image and time-series datasets and the results show the model outperforms its beaseline in accuracy of counterfactual estimation.

**Questions:**

Q1) Based on the experimental results (Figure 3) with the authors' argument, it's a bit hard to understand that the lack of guarantees on pre-existing deep structural equation models tend to generate potential incorrect reconstructions. The shown results are relatively weak to support their argument. Can authors show more convincing resutls on that?

Q2) The authors need to describe how their assumption on strong ignorability works on time-series data. The authors said they perform counterfactual estimation of treatment on patient's trajectory in which the observed variable Y can be time-wisely influenced by multiple treatments or other confounding factors. Considering the fact that the patient in a hospitcal (a general ward/ICU) usually recieves multiple treatments, I wonder how the authors handle this problem.

For improvement, the authors need to describe how they address counterfactual estimation on time-series data setting.

Q3) It's hard to see the benefit of spliting Uε to a categorical variables and continuous variables. Can authors describe more on that in detail?

Q4) In the experiment on cardiovascular simulator, what kind of dataset does the authors use? They should describe more explanations about the dataset and explain  how they preprocess data when handling clinical records (such as time points for fluid intake, length of sequence) and how a training set of patients are selected from the entire clinical database (e.g., the selected patients show normal range of blood pressure? or what types of other clinical events do these patients have in their EHRs?). Also, more detailed explanations about qualitative analysis on cardiovascular experiments will be helpful for showing the effectiveness of their method. For example, how clinical meanful is the estimated patient trajectory from counterfactuals?

**Limitations:**

The authors addressed the limitations.


**Strengths And Weaknesses:**

The tackled goal is an interesting topic in the field of counterfactual inference and estimation. The proposed methodology is well described and the quality of writings is good.

However, it's a bit hard to understand the benefit of proposed methodology and, although interesting, it’s hard to say the introduced technique is quite new as the proposed method seems to be a simple extension from pre-existing method (structural causal model). Also, the paper lacks detailed explanation and proof for their argument that pre-existing structural causal equation models fail to provide theoretical guarantees for reconstruction of counterfacturals. Although the authors show some qualitative result (Figure 3) for supporting their argument, it's not quite convincing. It will be helpful if the authors are able to provide more results and explanations for this.

Furthermore, even though the authors argue that the proposed method provides a counterfactual reconstruction architecture on images and time series, the experimental results do not seem to be good enough to show benefits of their method. Especially for time-series experiment, more diverse qualitative and quantitative experiments should be performed.

---

> ### Author Response · Authors · 2022-07-28
> **Answer to questions (1/3)**
>
> Dear Reviewer dSw5,
>
> Thank you for your constructive criticism and for the time taken to evaluate our work. Based on the points raised, it appears that some key points of our contributions were not fully clear and we hope the following will help in clarifying our work. Please let us know if any questions remain and we will be glad to address them in detail.
>
> ### About the positioning and novelty of our work
>
> >"However, it's a bit hard to understand the benefit of proposed methodology and, although interesting, it’s hard to say the introduced technique is quite new as the proposed method seems to be a simple extension from pre-existing method (structural causal model). Also, the paper lacks detailed explanation and proof for their argument that pre-existing structural causal equation models fail to provide theoretical guarantees for reconstruction of counterfacturals."
>
> **Answer** Our approach indeed builds upon the structural causal literature. However our goal is to break free from the very narrow assumptions made in structural causal models, that prevent us from using general deep learning architecture for counterfactual inference. As stated in our Section 2.2 (*General non-identifiability of counterfactuals*), counterfactual identification is in general not possible without knowledge of the structural equations. See for instance [Pearl2021 (definition2),Pearl2009,Oberst2019] for reference and discussion of the problem. The major assumption of structural equation models thus resides in the knowledge of these equations describing the causal relations in the data. However, in many case, the exact functions describing the links between variables is not known and we wish to infer them from data (e.g. with neural networks).  In this case, the structural causal model assumption fails to provide us with a viable solution. Our contribution lies in proposing counterfactual identifiability guarantees when the structural equations are not know (and thus learnt from data), assuming categorical backward variables. In that sense, *we are proposing a trade-off between structural equation models (identifiability but assumption of knowledge of the structural equations) and general deep learning architecture [Pawlowski2020] (structural equations learnable from data but no identifiability guarantees)*.
>
> We hope this clarifies how our model differs from both structural equation models and general deep learning architectures.
>
> **References**
>
> - Pearl, Judea. "7.1 Causal and Counterfactual Inference." The Handbook of Rationality (2021): 427.
> - Pearl, Judea. Causality: Models, Reasoning and Inference. Cambridge university press, 2009.
> - Oberst, Michael, and David Sontag. "Counterfactual off-policy evaluation with gumbel-max structural causal models." International Conference on Machine Learning. PMLR, 2019.
> - Pawlowski, Nick, Daniel Coelho de Castro, and Ben Glocker. "Deep structural causal models for tractable counterfactual inference." Advances in Neural Information Processing Systems 33 (2020): 857-869.

---

> > ### Author Response · Authors · 2022-07-28
> > **Answer to questions 2/3**
> >
> > ### About the relevance of our results
> > > Based on the experimental results (Figure 3) with the authors' argument, it's a bit hard to understand that the lack of guarantees on pre-existing deep structural equation models tend to generate potential incorrect reconstructions. The shown results are relatively weak to support their argument. Can authors show more convincing resutls on that?
> >
> > **Answer** Based upon the above discussion, our experiments section aims at showing that in practice this lack of identifiability leads to poorer counterfactual reconstructions. From Table 1, one can observe that other approaches fail to recover counterfactuals as accurately as our method. Our approach significantly outperforms all other methods on all datasets. As our goal is to predict counterfactuals accurately we think that evaluating the discrepancy (MSE and PEHE) between true and reconstructed counterfactuals is an informative metric. However, we are very willing to implement any other suggestion that you might have to improve the clarity of our point.
> >
> > ### About using our model for time series data
> > > The authors need to describe how their assumption on strong ignorability works on time-series data. The authors said they perform counterfactual estimation of treatment on patient's trajectory in which the observed variable Y can be time-wisely influenced by multiple treatments or other confounding factors. Considering the fact that the patient in a hospitcal (a general ward/ICU) usually recieves multiple treatments, I wonder how the authors handle this problem.
> >
> > **Answer** Our approach is amenable to time series counterfactuals estimation in the following way. Let’s consider each treatment assignment individually. That is, the counterfactual question focuses on a single temporal treatment assignment. For instance, “what would have happened if we had given this patient 100ml of that drug instead of 200ml?”. In this case, $X$ is the previous longitudinal information known about the patient up until the time of treatment (so $X$ is the whole time series up until treatment time, including previous treatments). As shown on Figure 1, we assume no hidden confounding, such that there is no other variable than $X$ that influences both the treatment assignment and the treatment outcome. This comes down to assuming that only the longitudinal information that is observed up until time of treatment is used by the clinician to assign the treatment. What is more, this instantiation of the ignorability assumption in time series has been used in previous works [Lim2018, Bica2020]. Our approach is thus inline with the literature regarding single treatment counterfactuals and our results suggest that we can recover time series counterfactuals.
> >
> >  When it comes to dynamic treatment regimes, that is, asking counterfactual questions with respect to multiple successive treatments (e.g. what would have happened if we would have followed treatment protocol A instead of treatment protocol B), our approach is not yet applicable. We believe that this more challenging case is an interesting future direction.
> >
> > Finally we want to stress that our approach is meant to be general in terms of counterfactual reconstructions (time series, images or other data modalities) and is not limited to time series.
> >
> > #### References
> > - Lim, Bryan. "Forecasting treatment responses over time using recurrent marginal structural networks." *advances in neural information processing systems*
> >  31 (2018).
> >
> > - Bica, Ioana, et al. "Estimating counterfactual treatment outcomes over time through adversarially balanced representations." *arXiv preprint arXiv:2002.04083*
> >  (2020).

---

> > ### Author Response · Authors · 2022-07-28
> > **Answer to questions 3/3**
> >
> > ### About splitting $U_{\epsilon}$ into $U_Z$ and $U_{\eta}$
> >
> > > It's hard to see the benefit of spliting Uε to a categorical variables and continuous variables. Can authors describe more on that in detail?
> > >
> >
> > **Answer** As detailed in our answer above regarding the positioning of our contribution, our goal is to relax the strict assumptions of structural causal models and investigate what identifiability guarantees remain. In our work, we investigate what guarantees can be given when the background variables are dominated by a categorical component. As stated above, in the general case of arbitrary and unknown structural equations and general backward variables distribution, the counterfactual queries are non-identifiable. Yet, as we shown, if $U_{\epsilon}$ can be expressed by a strong categorical component $U_Z$ and a minor continuous component $U_{\eta}$, then we can derive a probabilistic bound on the quality of the counterfactual reconstruction. Furthermore, one could consider $U_{\eta}$ to be trivially zero. However that would result in a stronger assumption for the generative model. We thus study the more general case where $U_{\eta}$ is non trivial.
> >
> > ### Details about the cardiovascular dataset
> >
> > > In the experiment on cardiovascular simulator, what kind of dataset does the authors use? They should describe more explanations about the dataset and explain how they preprocess data when handling clinical records (such as time points for fluid intake, length of sequence) and how a training set of patients are selected from the entire clinical database (e.g., the selected patients show normal range of blood pressure?
> > >
> >
> > **Answer** As we describe in Section 5.1.2 and in Appendix A.3, we use a cardio-vascular simulator as proposed in [Zenker2007]. We stress that because counterfactuals are by definition not observed, we cannot evaluate them on observational data. A widely accepted practice is thus to evaluate counterfactual (and individual treatment effects) on simulated but realistic data (See for instance [Bica2020,Shalit2017,Rui2020,DeBrouwer2022]). The generating equations are provided in details in Appendix A.3.
> >
> >
> > Zenker, Sven, Jonathan Rubin, and Gilles Clermont. "From inverse problems in mathematical physiology to quantitative differential diagnoses." *PLoS computational biology*
> >  3.11 (2007): e204.
> >
> > Bica, Ioana, et al. "Estimating counterfactual treatment outcomes over time through adversarially balanced representations." arXiv preprint arXiv:2002.04083
> >  (2020).
> >
> > Shalit, Uri, Fredrik D. Johansson, and David Sontag. "Estimating individual treatment effect: generalization bounds and algorithms." *International Conference on Machine Learning*
> > . PMLR, 2017.
> >
> > Li, Rui, et al. "G-Net: a deep learning approach to G-computation for counterfactual outcome prediction under dynamic treatment regimes." *arXiv preprint arXiv:2003.10551*
> >  (2020).
> >
> > De Brouwer, Edward, Javier Gonzalez, and Stephanie Hyland. "Predicting the impact of treatments over time with uncertainty aware neural differential equations." *International Conference on Artificial Intelligence and Statistics*
> > . PMLR, 2022.

---

> ### Author Response · Authors · 2022-08-09
> **Did we address all your concerns ?**
>
> Dear Reviewer dSw5,
>
> Thanks again for taking the time to review our paper and for your encouraging feedback ! As the discussion period is coming to a close, may we enquire if all the concerns you raised have been properly addressed ? Thank you very much,
>
> Best Regards,
>
> The authors

---

### Official Review · Reviewer_Md9L · 2022-07-12

**Rating:** 5
**Confidence:** 3
**Soundness:** 2 fair
**Presentation:** 2 fair
**Contribution:** 2 fair

**Summary:**

This paper proposes a counter-factual model with the assumption that the traditional exogenous variable is decomposed into independent categorical and continuous variable.
Here, only the categorical variable is governed by top-most exogenous variable, and due to the nature of categorical selection, the authors proposed clustering-based two-step counter-factual prediction model.



**Questions:**

- Is there any difference on the performance when using GMM instead of k-means clustering?
- Could you explain about Figure 3 more precisely? For example, the digit in the second row & the third column is drawn by intervening the color, am I correct? What is $T'$? Considering Figure 1, $U_z$ and $T$ are not linked. However, it seems that other factors, such as rotation (and probably also the noisy-ness, but not perfectly sure due to the low-resolution of MNIST data itself), have changed after intervening the color of digit in Figure 3, which is not desirable. What causes such phenomenon? Probably, the size of $T'$?
- $U_\eta$ can be some other factors. For example, let $T$ be some other underlying natures of digit data (ex. thickness, slope), and set $U_\eta$ as rotation (since it is enforced at data generation stage). I think it also can be one example of the causality. (Please let me know if I'm missing something.) Will the intervention cause the change of rotation in this case? What was the intuition behind of setting $U_\eta$ as noise?
- With the reconstructed counter-factual images, how does the digit classification result change? (with simple classifier)

**Limitations:**

The authors have discussed the limitations and potential negative societal impact in the main paper.

**Strengths And Weaknesses:**

- The paper is well-written and well-organized.
- Also, the quantitative results seems promising.
- The experiment is conducted two simple datasets and one real-world dataset. I would recommend conducting counter-factual image generation task on some other image dataset, perhaps CelebA, which can directly visualize and compare the effect and performance of proposed method against baselines.
- Setting $U_\eta$ as noise is a little bit unnatural. Regarding this, see questions below.

---

> ### Author Response · Authors · 2022-07-29
> **Answer to reviewer's comments (1/2)**
>
> Dear Reviewer Md9L,
>
> Thank you so much for your encouraging and insightful review. We are happy to report that we have implemented your suggestions about comparing a variation of our approach using GMM for the initial clustering step and we hope that the following comments with help answering your other questions about our work.
>
> ### About extra image datasets
> > I would recommend conducting counter-factual image generation task on some other image dataset, perhaps CelebA, which can directly visualize and compare the effect and performance of proposed method against baselines.
>
> - As we mention in Section 5.1 of the paper, counterfactual evaluation is challenging, because counterfactuals are, by definition, not observed. We thus follow the previous works here in evaluating our method on synthetic datasets (like [Pawlowski2020], which is the inspiration for the image experiment). That means that a dataset like CelebA cannot be readily used for counterfactual estimations as we cannot manipulate content of the image.
>
> ### About using GMM vs. K-means
> > Is there any difference on the performance when using GMM instead of k-means clustering?
>
> - Indeed, the *initial* clustering step of our algorithm described in Section 3.2 uses k-means but other clustering methods can be considered and explored. To investigate the impact of the clustering algorithm, we re-ran our model on all datasets with a Gaussian Mixture Model clustering step, as suggested. We report the results in Table 6 in the appendix. For convenience, we also report the results below. We observe that using GMM has a minor impact on performance, suggesting robustness of our architecture with respect to the initial clustering method. Reconstruction quality is improved for some datasets (e.g. Harmonic Oscillator) but degraded for others (e.g. Cardiovascular).
>
>  Counterfactual reconstruction (Test Mean Squared Error):
> | Model| Harmonic Oscillator (additive) | colored-MNIST (additive) | Cardiovascular (additive)  |   Harmonic Oscillator (non-additive) | colored-MNIST (non-additive) | Cardiovascular (non-additive)  |
> |---|---|---|---|---|---|---|
> CFQP-Kmeans | $ 0.013 \pm 0.001$ | $\mathbf{0.001 \pm 0.001}$ | $0.077 \pm 0.050$ | $ \mathbf{0.009\pm0.001} $ | $ \mathbf{0.002\pm0.001} $ | $ \mathbf{0.188\pm0.114} $
> CFQP-GMM | $\mathbf{0.001 \pm 0.001}$ | $0.002 \pm 0.001$ | $ \mathbf{0.067\pm0.042}$  | $\mathbf{0.009\pm0.001} $ | $\mathbf{0.002 \pm 0.001}$ | $ 0.192\pm0.0834 $

---

> > ### Author Response · Authors · 2022-07-29
> > **Answer to reviewer's comments (2/2)**
> >
> > ### About Figure 3.
> > > Could you explain about Figure 3 more precisely? For example, the digit in the second row & the third column is drawn by intervening the color, am I correct? What is ? Considering Figure 1, $U_z$ and $T$ are not linked. However, it seems that other factors, such as rotation (and probably also the noisy-ness, but not perfectly sure due to the low-resolution of MNIST data itself), have changed after intervening the color of digit in Figure 3, which is not desirable. What causes such phenomenon? Probably, the size of $T'$? $U_{\eta}$  can be some other factors. For example, let T  be some other underlying natures of digit data (ex. thickness, slope), and set $U_{\eta}$  as rotation (since it is enforced at data generation stage). I think it also can be one example of the causality. (Please let me know if I'm missing something.) Will the intervention cause the change of rotation in this case? What was the intuition behind of setting  as noise?
> >
> > - In this experiment, the intervention is the rotation angle of the digit. The observed vector (X) is the un-rotated image. The coloring is the additional categorical noise Uz that impacts the treatment response (it is the hidden group that will modulate the treatment outcome). Our goal here is to predict the counterfactual, i.e. what would have been the outcome of another treatment assignment, conditioned on an unrotated image, an observed treatment (rotation angle) and the corresponding outcome. If we use the analogy of clinical patient data, we are asking what would have been the effect of another treatment than the one that was actually given to particular patient, knowing what happened to that patient. In the patient case, other external unobserved factors influence the treatment response, such as for instance, the blood type of the patient (let’s assume it’s not directly observed). To answer the counterfactual question accurately, one then has to use the observed treatment response to infer the blood type of the patient and use it to predict the outcome at another treatment value. Using this analogy, the unrotated image represents the available information about the patient before treatment ($X$), the rotation corresponds to the treatment ($T$), the color corresponds to the blood type (the unobserved categorical variable $U_Z$), the observed rotated (and colored) digit corresponds to the treatment outcome ($Y_T$). Finally $U_{\eta}$ corresponds to any additional unobserved continuous exogenous variable that might also influence the treatment outcome (on top of $U_Z$). In the clinical case, it could be for instance the skills of the surgeon. In the image case, we model this term by adding noise to the pixel values. Crucially, our work shows that if the impact of $U_Z$ on the treatment outcome dominates the impact of $U_{\eta}$ (e.g. the blood type is a more important driver than the surgeon skills), we can still provide theoretical guarantees for counterfactual reconstructions. Building upon those results, what we are showing in Figure 3 are the counterfactual predictions based on the observed outcome for treatment $T$. We set different values of $T’$ and see that our method is the only one capable of faithfully reconstructing the counterfactuals.
> >
> > - Regarding the slight variations of the images in the first row, you are right than in theory they should be identical and only rotated (as only the treatment $T$ is changed and $U_Z$ and $U_{\eta}$ are kept fixed). But indeed, the pixel discretization of MNIST makes it appear not exactly identical, so it’s an artifact from the low resolution of the images.
> >
> > ### About the change in classification
> > > With the reconstructed counter-factual images, how does the digit classification result change? (with simple classifier)
> >
> > - Our goal is not to classify the digits but rather use a similar dataset as in [Pawlowski2020] to showcase our ability to reconstruct counterfactuals from images. As stated above, because counterfactuals are by default not observed, a careful evaluation with respect to ground truth can only be carried out with synthetic data. Please let us know if further clarification is needed.
> >
> > **References**
> >
> > Pawlowski, Nick, Daniel Coelho de Castro, and Ben Glocker. "Deep structural causal models for tractable counterfactual inference." *Advances in Neural Information Processing Systems*
> >  33 (2020): 857-869.

---

> ### Author Response · Authors · 2022-08-09
> **Did we address all your concerns ?**
>
> Dear Reviewer Md9L,
>
> Thanks again for taking the time to review our paper and for your encouraging feedback ! As the discussion period is coming to a close, may we enquire if all the concerns you raised have been properly addressed ? Thank you very much,
>
> Best Regards,
>
> The authors

---

### Author Response · Authors · 2022-08-01
**General comment to all reviewers**

Dear Reviewers,

We want to thank you all again for taking the time to review our work and provide insightful feedback. We have implemented all your requests in the new version of the paper and made our best to answer your questions as exhaustively as possible. From the comments you raised, we noticed some recurrent topics that we would like to address more generally in complement to the individual answers we have provided.

**About the general motivation**

Our main objective is to predict counterfactuals from observational data. This is incredibly useful to assess treatment effects *a posteriori*, such as in analyzing clinical trials data or evaluating the impact of policies. However, this is known to be impossible from data alone. Researchers have thus proposed assumptions under which counterfactual prediction is possible. The classical approach resides in assuming a specific structural equation model. However, this is extremely limiting as it pre-supposes the knowledge of the equations that generate the data. Indeed, one would wish to be able to learn these equations from data instead. For instance, to take a clinical example discussed in the paper, we might want to *learn* how fluid intake impacts arterial blood pressure rather than having to posit a specific response function.

Our contribution is to relax the structural model assumptions at the expense of allowing a small error in the counterfactual reconstruction. While previous work has focused on showing exact identifiability of the counterfactual queries, we propose to bound the error on the counterfactual reconstruction. This allows us to learn the response function from the data, under less strict assumption. In particular, we can then model treatment response between high dimensional objects such as time series or images and still provide some guarantees in terms of the counterfactual estimation.

We believe that exact identifiability is often not required in practice and that many applications could allow for some bounded error in the reconstruction. This is the avenue that our work is opening up. The set of assumptions proposed here is certainly not unique and we envision that other works could propose other assumptions going in that direction in the future.

**About the evaluation**

Counterfactuals are by definition not observed, which makes the evaluation challenging. To evaluate our approach, we have thus followed the common practice of using semi-synthetic datasets, for which ground-truth counterfactuals can be generated. This way, we can reliably evaluate our counterfactual predictions against the ground truth. This evaluation is widely used in causal inference in general, for individual or average treatment effects prediction as well as counterfactuals.

---

### Author Response · Authors · 2022-08-08
**We are available to answer any further comments reviewers might have**

Dear Reviewers,

We want to thank you again for reviewing our paper and providing insightful feedback. We hope our comments provide satisfactory answers to all your questions. If this is not the case, please let us know as soon as possible and we will be glad to provide you with complementary responses. In case you are satisfied with our responses, we would greatly appreciate if you can consider updating your score accordingly.

Thank you very much,

The authors

---

### Meta-Review · Area_Chair_rJxu · 2022-08-27

**Recommendation:** Accept
**Confidence:** Certain

**Metareview:**

The authors develop a technique for estimating counterfactuals. Counterfactuals are generally not identifiable. This paper makes assumptions on the exogenous noise on the outcome to estimate counterfactuals. Namely, the paper assumes a purely exogenous continuous part and a discrete part. The reviewers were generally positive with the one negative reviewer noting in discussion that they still had concerns about novelty, but were swayed positive by the value the method could have on real world problems.


**Award:**

No

---

### Decision · Program_Chairs · 2022-09-14

Accept